



# Modelling the potential impacts of the recent, unexpected increase in CFC-11 emissions on total column ozone recovery

James Keeble[1,2], N. Luke Abraham[1,2], Alexander T. Archibald[1,2], Martyn P. Chipperfield[3,4], Sandip Dhomse[3,4], Paul T. Griffiths[1,2], and John A. Pyle[1,2]

[1]Department of Chemistry, University of Cambridge, Cambridge, UK

[2]National Centre for Atmospheric Science (NCAS), University of Cambridge, Cambridge, UK

[3]School of Earth and Environment, University of Leeds, Leeds, U.K.

[4]National Centre for Earth Observation (NCEO), University of Leeds, Leeds, U.K.

*Correspondence to*: James Keeble (james.keeble@atm.ch.cam.ac.uk)

**Abstract.** The temporal evolution of long-lived, anthropogenic chlorofluorocarbons is a key control on the timing of total column ozone (TCO) recovery. Recent observations have shown that the atmospheric mixing ratio of CFC-11 is not declining as expected under complete compliance with the Montreal Protocol, and indicate a new source of CFC-11. In this study, the impact of a number of potential future CFC-11 emissions scenarios on TCO recovery is investigated using the UM-UKCA model. Key uncertainties related to this new CFC-11 source and their impact on the timing of TCO recovery are explored, including: the duration of new CFC-11 emissions/production; the impact of any newly created bank; and the effects of co-production of CFC-12. Scenario-independent relationships are identified between cumulative CFC emissions and the timing of ozone recovery, which can be used to establish the impact of future CFC emissions pathways on ozone recovery in the real world. It is found that, for every 200 Gg Cl emitted, the timing of global TCO recovery is delayed by ~0.56 years. However, a marked hemispheric asymmetry in the latitudinal impacts of cumulative Cl emissions on the timing of TCO recovery is identified, with longer delays in the southern hemisphere than the northern hemisphere for the same emission. Together, these results indicate that, if rapid action is taken to curb recently identified CFC-11 production then no significant delay in the timing of TCO recovery is expected, highlighting the importance of ongoing, long-term measurement efforts to inform the accountability phase of the Montreal Protocol. However, if the emissions are allowed to continue into the future, and are associated with the creation of large banks, then significant delays in the timing of TCO recovery may occur.



# 1 Introduction

The discovery of the ozone hole by Farman et al (1985) led quickly to the confirmation of the idea put forward by Molina and Rowland (1974) that chlorine radical species, the breakdown products of the chlorofluorocarbons (CFCs), could deplete stratospheric ozone. In the face of the scientific evidence, the Montreal Protocol on substances that deplete the ozone layer

was agreed in 1987 and ratified in 1989. The original controls proposed were modest, covering only $CFCl_3$ (CFC-11), $CF_2Cl_2$ (CFC-12), three further CFCs and three brominated compounds (Halons). However, in line with the developing scientific understanding, the controls were subsequently strengthened in a series of adjustments and amendments to the Protocol. These included stronger regulation on the phase down schedules, the addition of many more CFCs, $CCl_4$ and transitional hydrochlorofluorocarbon (HCFC) compounds under the London Amendment in 1990 and the inclusion of many more

brominated compounds, including methyl bromide under the Copenhagen Amendment of 1992. The initial replacements for the CFCs, the HCFCs, have shorter lifetimes than the CFCs (Chipperfield et al., 2014) and accordingly their impact on stratospheric ozone is less. They, in turn, are being replaced by hydrofluorocarbons (HFCs), compounds which do not directly lead to ozone depletion but some of which are strong greenhouse gases. Regulations to limit the growth of many HFCs were agreed in the Kigali Amendment in 2016.

In consequence, the atmospheric abundances of ozone-depleting chlorine and bromine species are now declining in the atmosphere, following their peaks in the late 1990s (WMO, 2018), leading to the start of ozone recovery. The 2018 Scientific Assessment for the Montreal Protocol (WMO, 2018) reported that the Antarctic ozone hole, while continuing to occur each year, is showing early signs of recovery (e.g. Solomon et al., 2016) and that upper stratospheric ozone has increased by up to 3% since 2000 (e.g. LOTUS, 2019). However, there is as yet no significant recovery trend in global column ozone (e.g. Weber

et al., 2018). Furthermore, because many of the ODSs are also greenhouse gases, their control has brought significant climate benefits (Velders et al., 2007; Velders et al., 2012; WMO, 2018). The annual reduction in these greenhouse gas emissions is estimated to be about five times larger than the annual emission reduction target for the first commitment period of the Kyoto Protocol (WMO, 2014).

The Montreal Protocol has undoubtedly been successful. Without the Protocol the abundance of ODS would likely have risen

such that, for example, very large ozone depletion could have occurred in the Arctic (Chipperfield et al, 2015). Uncontrolled growth of the ODSs would also have severely exacerbated the impact on global warming of the increase in other greenhouse gases, making current climate targets even more challenging to meet. However, there have recently been questions about the completeness of the implementation of the Protocol. The concentrations of some short-lived anthropogenic halocarbons, which are not covered by the Protocol, have increased in the atmosphere (Hossaini et al., 2017; Oram et al., 2017; Fang et al. 2018),

with suggestions that they might be by-products in the production of other halocarbons. Furthermore, the concentrations of some of the controlled ODSs have not followed projections based on their phase-out under the Montreal Protocol. For instance, concentrations of carbon tetrachloride, $CCl_4$, have not fallen as rapidly as expected based on its atmospheric lifetime. A detailed reanalysis of $CCl_4$ indicates that inadvertent by-product emissions from the production of chloromethanes and





perchloroethylene, and fugitive emissions from the chlor-alkali process, have contributed to this discrepancy (SPARC, 2016; Sherry et al., 2018; WMO, 2018) and recently Lunt et al. (2018) have shown that emissions of $CCl_4$ from east China have increased in the last decade. East China as a source of other short-lived halocarbons was also suggested by Ashfold et al. (2015) and Fang et al. (2018).

Against this background, Montzka et al. (2018) showed that the atmospheric abundance of one of the major chlorine-carrying CFCs, CFC-11, is not declining as expected under complete compliance with the Montreal Protocol. Using the NOAA network of ground-based observations, they demonstrated clearly that the rate of decline of CFC-11 in the atmosphere between 2015-2017 was about 50% slower than that observed during 2002–2012 and was also much slower than had been projected by WMO (2014). They inferred that emissions of CFC-11 had been approximately constant at ~55 Gg yr-1 between 2002 and 2012 and

had then risen after 2012 to ~68 Gg yr-1. Current emissions are about 35 Gg yr-1 greater than anticipated by the WMO (2014) A1 scenario and Montzka et al. (2018) argued that this increase could not be explained by increased release from pre-existing banks. Instead, they suggested that production of CFC-11 in east Asia, which is inconsistent with full compliance of the Montreal Protocol, was the likely cause. Using inverse modelling, Rigby et al. (2019) have now shown that the increase in CFC-11 emissions from east China between 2008-2012 and 2014-2017 is ~7 Gg yr-1, corresponding to approximately 40-60%

of the global emission increase identified by Montzka et al. (2018) during that period.

The exact source of the emissions remains unknown, nor is it known if there is co-production of CFC-12. It is usual that the gases are produced together in an industrial plant (e.g. Siegemund et al., 2000), with the fraction of CFC-11 to CFC-12 production varying between 0.3 to 0.7 (UNEP, 2018). There is currently no evidence that CFC-12 concentrations in the atmosphere are also declining at a slower rate than expected but some co-production of CFC-12 along with CFC-11 is always

expected.

Compliance with the Montreal Protocol is essential for its continued success in reducing stratospheric $Cl_y$ and ultimately healing the ozone layer. Any non-compliance will inevitably prolong the period when the Antarctic ozone hole will continue to occur and delay the recovery of global ozone. It is essential therefore to understand the likely impact of the current non-compliance. Here the UM-UKCA chemistry-climate model is used to assess the possible implications of the change in decline

of CFC-11. A number of possible scenarios are explored in a range of sensitivity calculations. These include emissions which cease immediately, or which persist for different periods into the future. These scenarios also consider that some of the non-compliant production may be stored in new CFC-11 banks, for later release, and, that production of CFC-11 may be associated with co-production of CFC-12.

Until the source of the recent CFC-11 emissions is understood and thoroughly quantified, model calculations can only

investigate a range of possible future emissions scenarios. However, models can be used to search for a relationship between the amount of chlorine emitted into the atmosphere and the timing of the recovery of total column ozone (TCO). Here results from both the UM-UKCA model and the TOMCAT CTM (recently used to study the impact of increased CFC-11 emissions on the behaviour of the Antarctic ozone hole (Dhomse et al., 2019)) are used to investigate the relationship between increased



emissions and enhanced ozone depletion. Identification of a robust relationship would allow us to develop a scenario-independent understanding of the impact of uncontrolled CFC emissions on the timing of TCO recovery.

In Section 2 the UM-UKCA model and CFC scenarios used in this study are discussed in detail. Section 3 assesses the impact of these CFC emissions scenarios on stratospheric chlorine loading, before the impacts on the timing of ozone recovery are

examined in Section 4. Section 5 investigates the relationship between cumulative CFC emissions and ozone depletion, using both the CCM and CTM. Finally, further discussion of the results and a summary of our conclusions are provided in Section 6.

## 2 Model configuration and simulations

To explore the impacts of potential future CFC-11 emission scenarios on total column ozone recovery, a total of 10 transient

simulations were performed using version 7.3 of the HadGEM3-A configuration of the Met Office's Unified Model (Hewitt et al., 2011) coupled with the United Kingdom Chemistry and Aerosol scheme (hereafter referred to as UM-UKCA). This configuration of the model has a horizontal resolution of 2.5° latitude × 3.75° longitude, and 60 vertical levels following a hybrid sigma-geometric height coordinate, with a model top at 84 km. The chemical scheme is an expansion of the scheme presented in Morgenstern et al. (2009) in which halogen source gases are considered explicitly and the effects of the solar

cycle are considered, as described in Bednarz et al. (2016). The version of UM-UKCA used in this study is an atmosphere-only configuration, with each simulation using the same prescribed sea surface temperatures (SSTs) and sea ice fields taken from a parent coupled atmosphere-ocean HadGEM2-ES integration. Further information on the model configuration used for this study is provided in Keeble et al. (2018). Except for CFC-11 and CFC-12 lower boundary conditions (LBCs), all other chemical forcings in the simulations follow the experimental design of the WCRP/SPARC CCMI REF-C2 experiment (Eyring

et al., 2013), which adopts the RCP6.0 scenario for future GHG and ODS emissions. A baseline experiment performed using CFC-11 and CFC-12 LBCs provided by the WMO (2014) A1 scenario was run from 1960 to 2099. A further 9 simulations were performed, running from 2012 to 2099, using a range of CFC-11 and CFC-12 LBCs, which were designed to cover a large but plausible range of potential future CFC emissions scenarios given the associated uncertainties.

### 2.1 CFC-11 scenarios

There are a number of important details associated with the recently reported CFC-11 emissions from East Asia which are poorly understood. A key factor is whether the identified CFC-11 emissions arise from emissive or non-emissive uses. If they arise from a totally emissive use, then the observed CFC-11 changes represent the total new CFC-11 production, with this new source of CFC-11 being released into the atmosphere during either production or use. Conversely, if they arise from a non-emissive use (e.g. foam insulation), then the observed changes to CFC-11 represent only a fraction of the total production,

with a large component entering a new bank. In order to address this, two scenario types were created, which reflect the new



CFC-11 production which is in addition to that implied in the WMO (2014) A1 scenario. In SCEN1, which represents the emissive use scenario, constant emissions of 35 Gg CFC-11 yr-1 (~27 Gg Cl yr-1) were assumed while in SCEN2 total production was assumed to be 90 Gg CFC-11 yr-1 (~70 Gg Cl yr-1), with 15 Gg CFC-11 yr-1 of this total directly emitted into the atmosphere, while the remaining 75 Gg CFC-11 yr-1 entered a bank with an assumed release fraction of 3.5 % yr-1. The

SCEN2 scenario is designed to give an additional emission increment of ~35 Gg CFC-11 yr-1 in 2019, similar to the emissions value used in SCEN1 and consistent with the CFC-11 emissions increment, in addition to that expect assuming only release from known banks, identified by Montzka et al. (2018.) We emphasis that the 35 Gg CFC-11 yr-1 and 90 Gg CFC-11 yr-1 values represent the extra emissions/production increment assumed in addition to those of the WMO (2014) A1 scenario.

A second key factor is the duration of the illegal production. In order to address this question, three sets of simulation were

performed for both SCEN1 and SCEN2, in which uncontrolled production either stops this year (2019), or continues into the future until 2027 or 2042, giving total emissions periods of 7, 15 or 30 years, respectively. All simulations assume that uncontrolled production and emission began in 2012. Simulations are named such that the scenario name is followed by the emission period, separated by an underscore (i.e. SCEN1_15 uses SCEN1 emissions from 2012 for 15 years).

A third consideration is the potential co-production of CFC-12. While there is no evidence currently that CFC-12

concentrations in the atmosphere are declining at a slower rate than expected (Montzka et al., 2018), CFC-12 is commonly co-produced alongside CFC-11. Accordingly, an additional scenario (SCEN3) was developed in which 90 Gg yr-1 of both CFC-11 and CFC-12 is produced. The same assumptions are made about the relative fraction entering the bank and the subsequent bank release rate as for SCEN2. SCEN3 was performed for the three different emissions periods used by SCEN1 and SCEN2, and simulations follow the same naming convention. As we consider both CFC-11 and CFC-12, all future emission/production

values will be given in Gg Cl, with 1 Gg CFC-11 equal to ~0.77 Gg Cl, and 1 Gg CFC-12 equal to ~0.59 Gg Cl.

The emissions for these various scenarios are shown (in Gg Cl) in Figure 1a, while Figure 1b shows the cumulative emissions and Figure 1c shows the size of the newly created bank. Note that SCEN3 scenarios include both CFC-11 and CFC-12, both scaled to Gg Cl and summed.

Figure 1a highlights that the SCEN1 scenario emissions stop at the end of the assumed production, while for the SCEN2 and

SCEN3 scenarios emissions continue throughout the 21st century long after the cessation of production due to the newly created bank. For SCEN2 and SCEN3 the shapes of the emissions curves are controlled by the combination of direct emission and newly created bank. While production continues, the yearly direct emissions remain constant, but the bank grows, leading to larger total emissions per year. At the moment production stops the direct emissions also stop, resulting in a marked step down in the emissions. After that point, all emissions emanate from the newly created bank. The maximum size of emission is

dictated by the duration of production. For comparison, Daniel et al. (2007) estimate that peak CFC-11 production, before the adoption of the Montreal Protocol, reached ~300 Gg Cl yr-1 in 1986-87. For SCEN1 scenarios, cumulative emissions increase only during the period of assumed production; there is no newly created bank. For the SCEN2 and SCEN3 scenarios cumulative emissions grow most rapidly during the period when direct emissions occur but they continue to increase throughout the 21st





century due to emissions from the bank. The size of the newly created bank is dependent on the duration of production and the release rate from the bank (Figure 1c), which was assumed to be 3.5% yr$_{-1}$.

As discussed above, the UM-UKCA model uses prescribed lower boundary condition (LBC) mixing ratios of CFCs. As a result, each of the emissions scenarios described above was converted into LBC mixing ratios using a simplified box model

which uses only the emissions flux and a CFC-11 lifetime of 55 years. This model reproduces to within good agreement the observed 1994-2017 CFC-11 surface mixing ratio when initialized with the 1994 values and using the emission estimates of Montzka et al. (2018). The time variation of CFC-11 at the surface is shown in Figure 2 for the different scenarios.

## 3 Stratospheric Chlorine

Increases in stratospheric chlorine will lead to ozone depletion, and so uncontrolled production of CFCs could obviously pose

a serious threat to the continued success of the Montreal Protocol. Modelled 40 km stratospheric inorganic chlorine (Cl$_y$) mixing ratios, averaged from 10°S-10°N, are shown in Figure 3, and stratospheric Cl$_y$ return dates (the date at which Cl$_y$ mixing ratios, averaged from 10°S-10°N at 40 km, return to the BASE 1980 value) are given in table 1. In the BASE simulation, stratospheric Cl$_y$ mixing ratio is projected to return to its 1980 value by 2058. Only small differences in the stratospheric Cl$_y$ return date are modelled for the SCEN1 simulations, with a maximum delay of 3 years occurring in the

SCEN1_30 simulation, which assumes the longest duration of additional CFC-11 emissions. However, large delays in the stratospheric chlorine return date are modelled in the SCEN2 simulations, which assume a large bank is also being produced alongside the direct atmospheric emissions (see figure 1). In the SCEN2_7 scenario, which assumes CFC-11 production stops in 2019, the stratospheric Cl$_y$ return date is delayed by 2 years, and for SCEN2_30, which assumes CFC-11 production continues until 2042, the stratospheric Cl$_y$ return date is delayed by 8 years. The delays highlight the potential importance for

ozone depletion of any bank produced and its subsequent emission into the atmosphere. The delay in stratospheric Cl$_y$ return dates is larger still if co-production of CFC-12 is considered, with the stratospheric Cl$_y$ return date being delayed by 14 years in the SCEN3_30 scenario considered here.

## 4 Modelled total column ozone response

Figure 4 shows the annual mean total column ozone (TCO) data from the BASE simulation (grey line) from 1960-2100,

averaged over 60°S-60°N. Consistent with previous studies, TCO values decrease sharply from 1980 to the late 1990s as a result of increasing stratospheric chlorine loadings, before gradually increasing throughout the 21$_{st}$ century. Superimposed on these long-term trends is an 11-year oscillation resulting from the solar cycle. Observed annual mean TCO values from version 2.8 of the Bodeker Scientific total column ozone dataset (Bodeker et al., 2005) are shown in purple. There is generally good agreement between modelled TCO values and the Bodeker dataset; decadal total column ozone changes, the response of





column ozone to the solar cycle and the magnitude of interannual variability are all well captured by the model throughout the time period during which the observations and model data overlap.

As discussed by Keeble et al. (2018; following WMO, 2007; Chipperfield et al., 2017), three stages of ozone recovery can be identified: (i) a slowed rate of decline and the date of minimum column ozone, (ii) the identification of significant positive

trends and (iii) a return to historic values. Here, we focus on the impact of uncontrolled CFC-11 emissions on the last of these metrics: return to historic values, with the historic baseline value defined as the total column ozone average from 1960-1980. This definition of the baseline period was chosen to avoid any sensitivity of the timing of TCO recovery to the choice of any individual year (which may be anomalously high or low). In actuality, global (60°S-60°N) TCO averaged from 1960-1980 in the BASE simulation is 298.1 DU, and the values for 1960 and 1980 are 297.5 DU and 298.0 DU respectively, and so there is

little difference between the timing of global TCO recovery to these three values. However, differences do occur in regions with high interannual variability (e.g. the Arctic).

As discussed by Keeble et al. (2018), the identification of total column ozone recovery is complicated by the effects of interannual variability. To mitigate these impacts, the effects of natural processes (such as volcanic eruptions, the QBO, ENSO and solar cycle) can be removed from the data using statistical techniques (such as Multiple Linear Regression, e.g. Keeble et

al., 2018), or the data can be smoothed by averaging across multiple years (e.g. Dhomse et al., 2018). Following Dhomse et al. (2018), annual mean UM-UKCA data is smoothed using an 11-point boxcar smoothing to reduce both the effects of short-term variability and the signature of the 11-year solar cycle. The smoothed data are shown on figure 4 as the black line. For the analysis in subsequent sections, all data from the BASE and SCEN simulations are smoothed using this method.

Ideally, to provide uncertainty estimates for our various integrations, a multi-member ensemble would be run for the BASE

and each SCEN calculation. However, in order to explore the largest possible range of future CFC-11 emissions scenarios, only one integration was performed for each scenario. Instead, data from a separate 5-member ensemble of 1980-2080 transient simulations (see Bednarz et al, 2016) is used to provide a rough estimate of the uncertainty range of recovery dates. These simulations used the same model configuration as used for our BASE and SCEN calculations but were forced with the older WMO (2011) CFC-11 and CFC-12 LBCs. Uncertainty estimates were estimated simply by identifying the ensemble member

with the largest difference in recovery date from the ensemble mean, and defining this difference as the uncertainty range (e.g. for 5 ensemble members where the return dates might be 2061, 2062, 2060, 2061 and 2066, the mean return date would be 2062 and the largest deviation is 2066, so the uncertainty estimate would be ±4 years). This provides an approximate indication of uncertainty for all latitude ranges considered here, except for the latitude range 90°S-60°S, where recovery occurred after 2080, and so lies outside the range of the ensemble.

For the BASE integration under the WMO (2014) scenario, a global (60°S-60°N) TCO return date of 2054±2 years was calculated, with the return dates for other latitudes shown in Table 1. In the next section, the impact of CFC-11 and CFC-12 emissions from the different SCEN scenarios on these recovery dates is assessed.



### 4.1 Global total column ozone

Figure 5 shows smoothed TCO values for the BASE and SCEN simulations, averaged from 60°S-60°N. All simulations show return to the baseline period of the 1960-1980 average between 2054 and 2064 (return dates provided in Table 1). The BASE simulation, which adopts the WMO (2014) LBC for ODSs and as such assumes the lowest anthropogenic $Cl_y$ emissions, recovers the earliest, in 2054, as discussed above. The impact of the additional CFC-11 and CFC-12 production scenarios investigated here is to delay the timing of this recovery. For SCEN1_7 and SCEN1_15, the delay is small and within the range of return dates calculated from the 5-member ensemble. Only SCEN1_30 of the SCEN1 scenarios shows a significant delay in return date of 3 years. In contrast, for the SCEN2 simulations, which assume the creation of a new bank and subsequent emissions of CFC-11 from that bank, substantial delay in the recovery of global TCO is modelled in both the SCEN2_15 and SCEN2_30 simulations (4 and 7 years, respectively). Only in the case that the assumed production of 90 Gg/yr stops this year (2019) is no significant delay in the return of TCO values to the baseline period modelled. The SCEN3 scenarios, which assume the co-production of CFC-12 alongside CFC-11, all show larger delays in the return dates, being 10 years for SCEN3_30.

### 4.2 Regional total column ozone

Regional TCO projections for the BASE and SCEN simulations are also shown in figure 5, from the Antarctic to the Arctic, and the timing of TCO recovery for each region is given in Table 1. Annual mean TCO values over Antarctica (90°S-60°S) recover to the 1960-1980 average by 2084 in the BASE simulation, 30 years after the global (60°S-60°N) TCO average is expected to recover. Substantial further delays in the date of Antarctic ozone recovery are modelled for a number of the SCEN simulations. Again, the impact of the SCEN1 scenarios, which assumes no newly created bank, is modest with no delay modelled, except for SCEN1_30, which returns to the 1960-1980 average in 2088. In contrast, large delays are modelled for both the SCEN2_15 and SCEN2_30 simulations, which have projected recovery dates of 2093 and 2095 respectively. If co-production of CFC-12 is considered, SCEN3_15 and SCEN3_30 suggest that 11-year averaged TCO values will not recover to the 1960-1980 baseline period by the end of the 21st century.

In the SH midlatitudes (60°S-30°S), delays in recovery date are modelled for a number of SCEN simulations. If production stops in 2019 there is essentially no delay, while scenarios with higher emissions or longer duration lead to delays between 6 (SCEN2) and 9 years (SCEN3_30, which includes co-production of CFC-12).

While most SCEN simulations project a delay in the recovery of Antarctic and SH midlatitude annual mean TCO, SCEN1_7, SCEN1_15 and SCEN3_7 all have earlier recovery dates than the BASE simulation. In the case of SCEN1_7 these changes in the SH midlatitudes are outside the model range calculated from the Bednarz et al (2016) 5-member ensemble, occurring 4 years earlier than in the BASE simulation. This may be because the uncertainty estimates calculated here from the 5-member ensemble do not fully capture the true system uncertainty, or that atmospheric chemistry-climate feedbacks may result in





increased TCO values in some locations despite increased stratospheric ODS values. For example, Keeble et al. (2014) show modelled wintertime TCO increases in the northern midlatitudes resulting from increased polar ozone depletion and associated changes in the lower branch of the BDC.

The observed ozone loss in the tropics has been small and, furthermore, future changes in the tropics are driven both by reductions in the stratospheric abundance of halogens, which tend to increase ozone, and the strengthening of the Brewer-Dobson circulation, which tends to decrease column ozone (e.g. Meul et al., 2014; Keeble et al., 2017). Here, tropical (30°S-30°N) TCO values are projected to recover by 2057 in the BASE simulation, and all SCEN simulations show significant delays to this recovery date except for SCEN1_7 and SCEN2_7 (i.e. those simulations which assume that uncontrolled production of CFC-11 stops in 2019, and there is no co-production of CFC-12). While TCO values are projected to return to the 1960-1980 average for the broad definition of the tropics used here, if a narrower definition is used (e.g. 5°S-5°N), TCO values do not recover to the 1960-1980 average at any point in the 21st century. This is consistent with the impacts on lower stratospheric ozone of the increased speed of the BDC resulting from anthropogenic GHG changes (e.g. Eyring et al., 2013; Meul et al., 2016; Keeble et al., 2017).

In the NH midlatitudes (30°N-60°N), TCO under the BASE projection is modelled to recover in 2047, considerably earlier than the SH midlatitudes. As at other latitudes, significant delay in recovery is modelled in the majority of SCEN simulations. In the Arctic (60°N-90°N), annual mean TCO values are projected to recover in the BASE simulation in 2048, again substantially earlier than the Antarctic recovery date. While significant delays for Arctic ozone are modelled in the majority of SCEN simulations, unlike at other latitude ranges, the latest recovery dates are not associated with the SCEN3 simulations, which assume co-production of CFC-12. Instead, the latest recovery date of 2057 occurs in the SCEN2_30 simulations. We ascribe this to the very high dynamical variability of the Arctic polar vortex, its subsequent impact on total column ozone, and the large uncertainties in defining recovery dates in this region. Bednarz et al. (2016), also using the UM-UKCA model, showed that, although springtime Arctic ozone was projected to return to 1980 values by the late 2030s, episodes of dynamically-driven very low ozone could be found well into the second half of this century, consistent with our annual mean results presented here.

## 5 Identifying scenario-independent relationships between CFC emissions and TCO recovery

While the SCEN simulations used in this study were designed to cover a broad range of potential future CFC-11 production scenarios, it is unlikely that future CFC-11 emissions will follow any of the scenarios described here. Therefore, we aim here to identify scenario-independent relationships between future CFC emissions pathways and the impact on TCO. For example, Dhomse et al. (2018) found relationships between chlorine return dates and a number of indicators of ozone recovery for a range of models (see, e.g., their Figure 8). In this study this relationship is further explored by linking TCO differences to emissions. Three emerging relationships are explored in the following sections: i) the timing of global TCO recovery as a function of $Cl_y$ return date; ii) the magnitude of annual mean TCO depletion in a year as a function of the cumulative CFC



emissions up to that year; and iii) the date of TCO recovery as a function of the cumulative additional CFC emissions by the end of the simulation.

## 5.1 Cl$_y$ return date vs timing of TCO recovery

The future evolution of stratospheric ozone mixing ratios follows closely the evolution of stratospheric Cl$_y$ (e.g. WMO, 2018).
Dhomse et al. (2018) found correlations between modelled return dates of stratospheric chlorine and ozone recovery dates for Antarctic and Arctic spring across a range of CCMI models. A similar relationship between the timing of Cl$_y$ recovery and the timing of global TCO recovery is identified in the SCEN simulations performed as part of this study, shown in Figure 6. As discussed above, the timing of Cl$_y$ recovery is defined as the date at which Cl$_y$ mixing ratios at 40 km return, averaged from 10°S-10°N, to their 1980s value. The relationship between the timing of global TCO recovery and date of Cl$_y$ recovery is
robust, with an r$_2$ of 0.92, and indicates that for every year Cl$_y$ recovery is delayed, the timing of TCO recovery is delayed by 0.64 years (given by the gradient of the linear fit through the points). The date of Cl$_y$ recovery itself is strongly linked to the assumed emissions. A robust linear relationship, with an r$_2$ value of 0.96, was identified between the total cumulative additional Cl emissions and the delay in Cl$_y$ return date. This relationship indicates that, for every additional 200 Gg of Cl (258 Gg CFC-11 equivalent) emitted by 2099 above those implied in the WMO (2014) scenario, Cl$_y$ return dates are delayed
by 0.86 years. It should be noted that the date of Cl$_y$ recovery occurs ~5 years later on average than the timing of global TCO recovery in the BASE and SCEN simulations (see Table 1), indicating that even after the time TCO values have returned to the 1960-1980 average, stratospheric chlorine mixing ratios remain elevated.

## 5.2 TCO depletion vs cumulative emissions

In order to explore the magnitude of annual mean TCO depletion in any year as a function of the cumulative Cl emissions up
to that year, results from the UM-UKCA SCEN simulations are supplemented by simulations performed with the TOMCAT chemistry transport model (CTM; Chipperfield et al., 2017). Both models have full stratospheric chemistry schemes but are independent of one another. The control CTM simulation (CTM_C) was performed until 2080 with repeating year 2000 meteorology and time-dependent future source gas surface mixing ratios. Two further simulations (described in detail in Dhomse et al. (2019)) were performed with additional future CFC-11 emissions (i) at constant 67 Gg yr$_{-1}$ (CTM_Fix) and (ii)
including simulation of a bank and production decreasing to zero over 10 years (CTM_Bank). Note that simulation CTM_Bank gives larger emissions than CFM_Fix until about the year 2040.
Figure 7 shows, for both the UM-UKCA and TOMCAT models, the cumulative additional Cl emissions plotted against the additional TCO depletion driven by the increased emissions compared with a base integration, averaged over 60°S-60°N. UM-UKCA values are calculated as the difference between each SCEN simulation with respect to the BASE simulation, while
TOMCAT values are calculated as CTM_Fix - CTM_C and CTM_Bank - CTM_C.





The estimated TCO depletion from both UM-UKCA and CTM simulations follows a reasonably compact, linear relationship with cumulative emissions. The CTM simulations use identical meteorology, without any chemistry-climate feedbacks, and so a comparison of two simulations gives a direct measure of the chemical ozone impact of the additional emissions. In contrast, UM-UKCA generates consistent time-varying meteorology for each of the SCEN simulations and so includes both a forced

response and internally generated natural variability. Despite this difference, the modelled relationship between cumulative emissions and global ozone depletion is remarkably similar for the two models: a total emission of 3500 Gg (Cl) gives a near-global mean ozone decrease of between 2.5-3 DU. The agreement between the two models provides confidence in the diagnosed chemical impact of the CFC-11 emissions and shows a means by which results from different model simulations, with different CFC-11 emissions scenarios, can be compared. Further, it allows an estimate to be made of the impact of any

given CFC-11 emission scenario on TCO depletion, which can in turn be applied to the real-world future emissions of CFC-11, which remain uncertain.

Note that in CTM run CFC_Bank the maximum CFC-11 emissions by 2080 are around 2700 Gg CFC-11. By this stage a significant part of the large initial pulse of emissions (see Dhomse et al 2019) would have been removed from the atmosphere (consistent with its ~55 year lifetime) so that the CTM line deviates from the simple linear fit (black line Figure 7). This is

also true for the SCEN simulations performed with UM-UKCA, although the effects are small compared to the dynamically driven variability. Note that the SCEN1 simulations are not shown in Figure 7 as their cumulative emissions stop increasing when the direct emissions stop (see Figure 1) and so for a large period of the simulations the differences are dominated by variability.

**5.3 Timing of TCO recovery vs total cumulative emissions**

Figure 8 shows the relationship between the total cumulative additional Cl emissions, in Gg, and projected global TCO recovery date in the BASE and SCEN simulations. Note that these values represent the additional Cl emissions assumed for each scenario in addition to the WMO (2014) A1 scenario. A robust ($r_2$=0.93) linear relationship is found, indicating that for every extra 200 Gg of Cl emitted, global TCO recovery is delayed by 0.56 years. Repeating this analysis for 10-degree latitude bins gives the latitudinal dependence of the impacts of cumulative Cl emissions on TCO recovery (Figure 9). Uncertainty

estimates are calculated for the regression using the standard error of the estimate, given as $\sigma_{est} = \sqrt{\frac{\Sigma(y-y')^2}{N}}$ where $y$ is date of TCO recovery, $y'$ is the predicted date of TCO recovery from the regression mode, and $N$ is number of simulations. For every 200 Gg of Cl emitted, TCO recovery is delayed by between 0.18-1.60 years, with a marked hemispheric asymmetry evident in the response. Large delays (~1.6 years per 200 Gg Cl) are modelled in the Antarctic, where heterogeneous processing of chlorine reservoirs on polar stratospheric cloud (PSC) surfaces allows for large ozone depletion even for relatively small

chlorine concentrations. On annual mean timescales these low values mix into the southern hemisphere midlatitudes, resulting in larger delays to recovery there (0.5-0.8 years per 200 Gg Cl) than in the northern hemisphere midlatitudes and Arctic (0.1-





0.5 years per 200 Gg Cl), where the effects of PSC processing are less pronounced due to the higher temperatures and high dynamical variability of the Arctic polar vortex. Note that no values are given south of 80°S, as TCO values do not recover to the baseline period by the end of the simulations.

The delay in tropical TCO recovery is also long (0.5-1.3 years per 200 Gg Cl) and associated with large uncertainties. As discussed above, the observed depletion of ozone in the tropics has been very small (WMO, 2018) and quantifying TCO recovery is complex, depending on not just the ODS loading of the stratosphere but on other factors including the future levels of greenhouse gases, changes to tropospheric ozone and the projected acceleration of the Brewer-Dobson circulation (e.g. Keeble et al., 2017).

## 6 Discussion and conclusions

The Montreal Protocol has been successful in reducing emissions of ODSs into the atmosphere, which in turn has led to the onset of ozone recovery. However, recent observational evidence indicates that atmospheric mixing ratios of CFC-11 are declining at a slower rate than that expected under full compliance with the Montreal Protocol. It seems likely that emissions resulting from new production, in contravention of the Montreal Protocol, are the likely cause of the change in decline rate, with an important source in east Asia (Montzka et al., 2018; Rigby et al., 2019). However, there remain large uncertainties associated with these emissions: their source remains unidentified, changes to the release rate from the historical bank are unknown, the size of any newly created bank is uncertain and undetected co-production of other chlorinated ODSs is possible. Given these uncertainties, the impact of a range of plausible future CFC-11 emissions scenarios on the timing of TCO recovery was explored using the UM-UKCA model. Making a range of assumptions the scenarios are intended to cover a breadth of future emission pathways. We consider the size of emissions and their duration; we compare emissive versus non-emissive use (where in the latter the bank is enhanced) and we consider possible co-production of CFC-12. While none of the scenarios developed here is expected to accurately predict the future CFC-11 emissions pathway of the real-world, they provide a sensitivity range to guide possible future trajectories of the ozone layer.

If the recently identified CFC-11 emissions result from an emissive use (i.e. there is no new bank being created and estimated emissions are equal to the total production) then, provided the source of new CFC-11 production stops within the next decade, results from the SCEN1 scenarios indicate that there will be no significant delay in the return of global total column ozone to the 1960-1980 baseline. Only in the case of prolonged emissions would significant delay in the recovery of global column ozone be expected.

However, if the recently identified CFC-11 changes result from non-emissive use, results from the SCEN2 scenarios indicate that, unless stopped immediately, the production has the potential to delay global total column ozone recovery by up to 7 years, depending on the duration of the production and the size of the annual increment to the bank. Further, results from the SCEN3 scenarios suggest that if CFC-12 has been co-produced with CFC-11, then global total column ozone return could have already





been delayed by 4 years, and that if, under the assumptions made here, production continues for up to 30 years from 2012, total column ozone recovery may be delayed by a decade.

Our results are, of course, dependent on the assumptions made in each of the SCEN scenarios. Therefore, it is important to identify scenario-independent metrics which can be used to establish the impact of future CFC emissions pathways on TCO

recovery. Three such relationships were identified: i) the timing of global TCO recovery as a function of $Cl_y$ return date; ii) the magnitude of annual mean TCO depletion in a year as a function of the cumulative CFC emissions up to that year; and iii) the date of TCO recovery as a function of the total cumulative additional CFC emissions by the end of the simulation. The second of these relationships was further verified by comparing with results from the TOMCAT CTM, and despite differences between the assumed emissions scenarios used by both models, and the fundamental differences in the treatment of

meteorology, good agreement was found between the two models, with 2.5-3 DU TCO depletion occurring for an additional 3500 Gg Cl. The robust linear relationship found between the total cumulative additional Cl emissions and timing of global TCO recovery indicates that for every 200 Gg of Cl (~258 Gg CFC-11) emitted, global TCO recovery is delayed by 0.56 years. However, a marked hemispheric asymmetry in the impacts of cumulative Cl emissions on the timing of TCO recovery at particular latitudes was identified, with longer delays in the southern hemisphere than the northern hemisphere for the same

emission.

While these scenario-independent relationships are useful, they come with some caveats. All the scenarios developed in this study assume that new CFC-11 production began in 2012, and that despite new CFC-11 production, atmospheric CFC-11 mixing ratios continue to decline, consistent with the observations presented by Montzka et al. (2018). We explore in this study uncertainties related to this new CFC-11 source. However, it is expected that 200 Gg of CFC-11 emitted in 2020 would

not have the same impact on ozone recovery as 200 Gg of CFC-11 emitted in, for example, 2080. Therefore, the temporal evolution of CFC-11 emissions is likely a key control on the relationships identified in this study. We have assumed steady changes in emissions, consistent with a continuous anthropogenic source of additional CFC-11, rather than changes which might be large but sporadic. As such, while the relationships identified here likely give a good indication of the TCO response to the recently identified source of CFC-11, they may not prove robust for any unexpected CFC-11 emissions later in the

century.

The detection of the change in the rate at which CFC-11 concentrations are declining in the atmosphere, and the inferred change in emissions, are important contributions to the Montreal Protocol during its accountability phase, during which the impact of the Protocol on the atmosphere is being assessed. It is clear that long-term monitoring of ODSs, as well as ozone, is an absolutely critical component of the atmospheric science response to the Protocol and its input to policy negotiations.

Continued modelling of the impact of these emissions on the projected timing of TCO recovery is also required.

Results presented here highlight the need for rapid action in tackling any uncontrolled production of CFC-11. Unless emissions are stopped rapidly, we anticipate potentially significant delays in recovery. Recovery of global ozone could be delayed by about a decade, on the basis of our assumed emissions, and Antarctic ozone might not recover at all this century. New



knowledge concerning the nature of the ODS emissions is required, which, in concert with increased atmospheric measurements of the ODSs, can inform the on-going discussions of the Montreal Protocol and ensure its future success.

**Data availability**

Data from all simulations are available on the UK Research Data Facility (http://www.archer.ac.uk/documentation/rdf-guide).
v2.8 of the Bodeker dataset can be found at http://www.bodekerscientific.com/data/total-column-ozone.

**Competing interests**

The authors declare that they have no conflict of interest.

**Acknowledgements**

The research leading to these results has received funding from the European Community's Seventh Framework Programme
(FP7/2007-2013) under grant agreement no. 603557 (StratoClim). We thank NERC through NCAS for financial support and NCAS-CMS for modelling support. Model simulations have been performed using the ARCHER UK National Supercomputing Service. This work used the UK Research Data Facility (http://www.archer.ac.uk/documentation/rdf-guide). We would like to thank Greg Bodeker of Bodeker Scientific, funded by the New Zealand Deep South National Science Challenge, for providing the combined total column ozone database.

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



| Simulation | Date of Cl$_y$ recovery | Date of total column ozone (TCO) recovery | | | | | |
|---|---|---|---|---|---|---|---|
| | 60°S-60° | 60°S-60° | 90°S-60°S | 60°S-30°S | 30°S-30°N | 30°N-60°N | 60°N-90°N |
| Base | 2058 | 2054±2 | 2084±? | 2065±1 | 2057±3 | 2047±2 | 2048±3 |
| SCEN1_7 | 2059 | 2055 | 2078 | 2061 | 2060 | 2050 | 2050 |
| SCEN1_15 | 2057 | 2055 | 2082 | 2064 | 2062 | 2050 | 2052 |
| SCEN1_30 | 2061 | 2057 | 2088 | 2068 | 2063 | 2049 | 2048 |
| SCEN2_7 | 2060 | 2055 | 2084 | 2065 | 2060 | 2047 | 2049 |
| SCEN2_15 | 2062 | 2058 | 2093 | 2071 | 2065 | 2051 | 2052 |
| SCEN2_30 | 2066 | 2061 | 2095 | 2071 | 2069 | 2052 | 2057 |
| SCEN3_7 | 2061 | 2058 | 2081 | 2064 | 2063 | 2052 | 2054 |
| SCEN3_15 | 2064 | 2058 | No recovery | 2068 | 2066 | 2053 | 2053 |
| SCEN3_30 | 2073 | 2064 | No recovery | 2074 | 2081 | 2052 | 2052 |

**Table 1** Dates of Cl$_y$ recovery (defined as the date at which Cl$_y$ mixing ratios at 40 km return to the BASE simulation 1980 value, averaged from 10°S-10°N) and TCO recovery (defined as the date at which TCO values return to the 1960-1980 average) for the BASE and SCEN simulations. Uncertainty estimates for TCO recovery in the BASE simulation are provided by a separate 5-member ensemble, as described in the text. TCO recovery occurs after 1980 for the latitude range 90°S-60°S, and so no uncertainty estimate can be provided for this latitude band, denoted by the '±?' in the table.





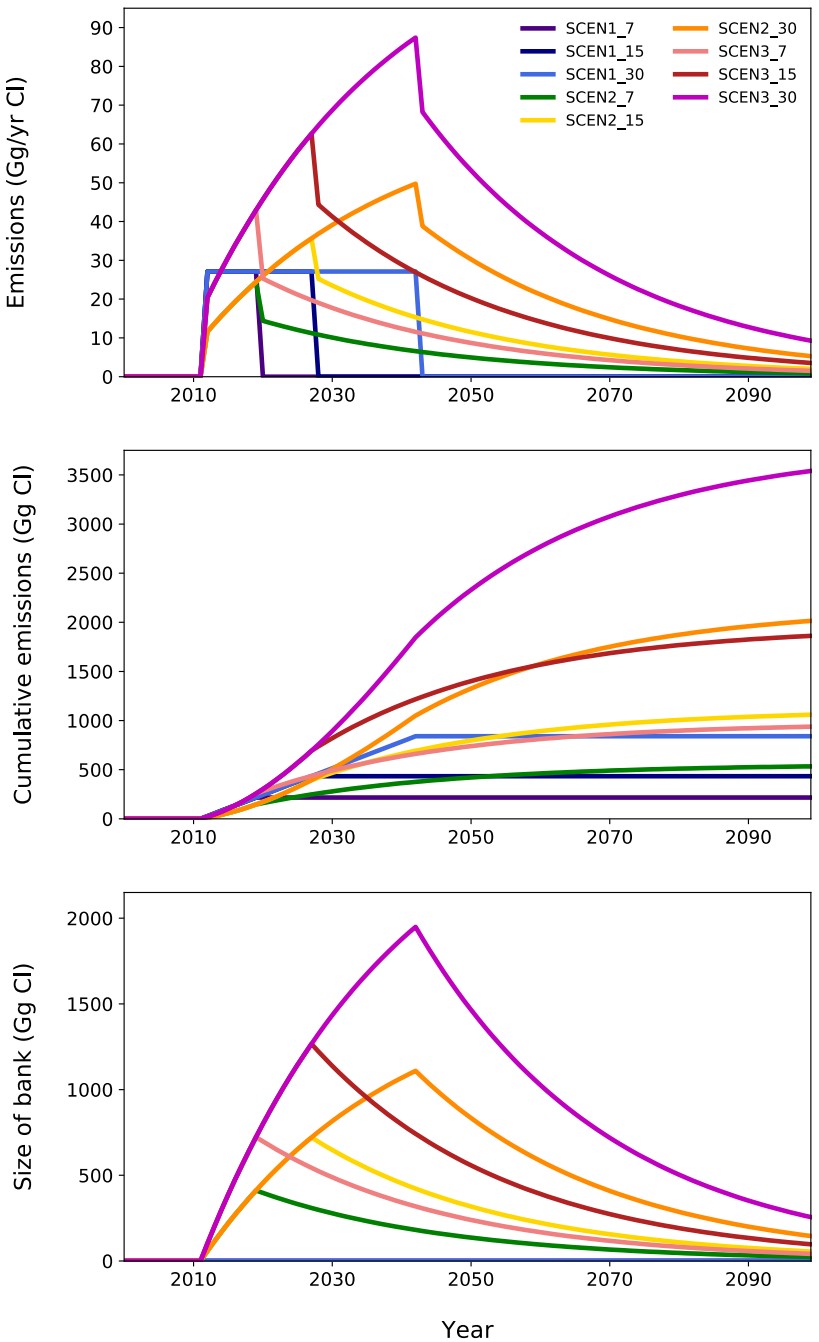

**Figure 1** (a) Additional emissions (in Gg Cl yr$^{-1}$) assumed for each of the SCEN simulations on top of those already assumed by the WMO (2014) A1 scenario. For all scenarios, the additional, uncontrolled source of emissions is assumed to start in 2012. (b) Cumulative additional emissions for each of the SCEN simulations. (c) The size of the newly created bank for the SCEN2 and SCEN3 scenarios.



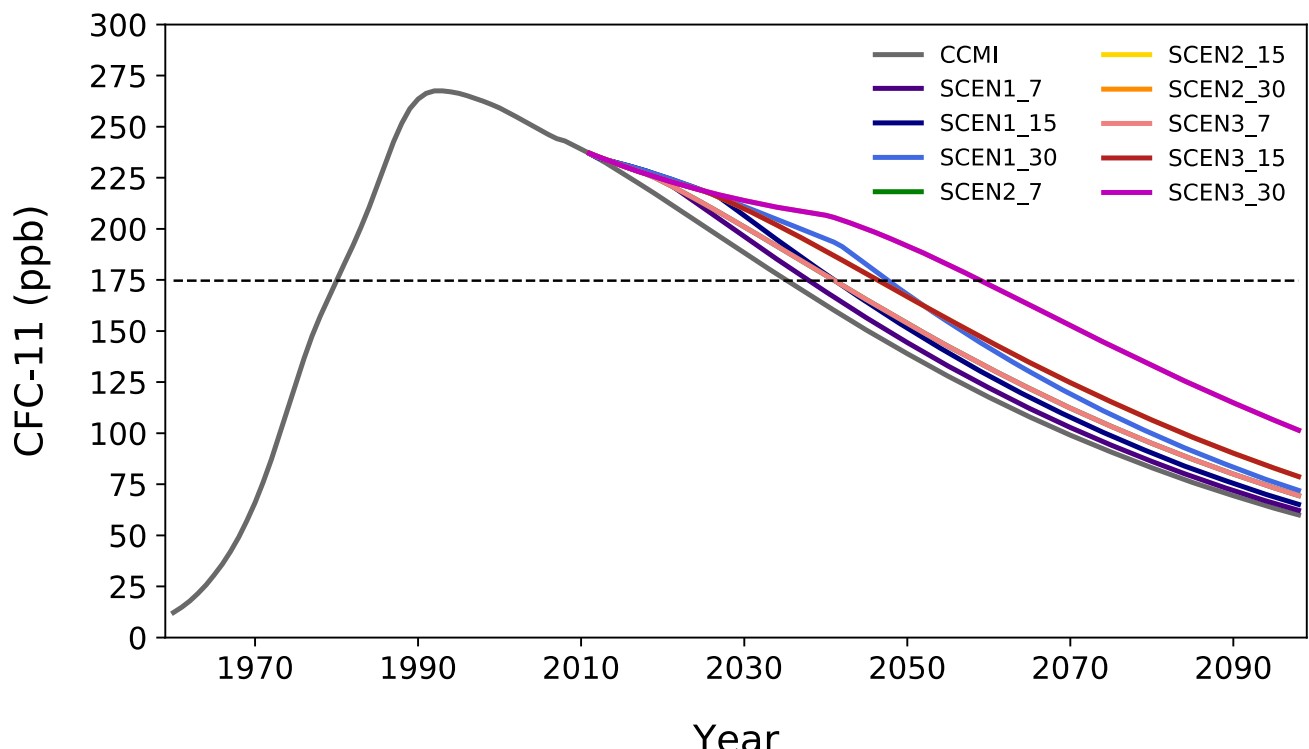

**Figure 2** Prescribed global mean surface lower boundary mixing ratio of CFC-11 used for the BASE and SCEN simulations. The dashed line represents the 1980 CFC-11 surface mixing ratio in the BASE simulation.  Note that the SCEN2 and SCEN3 simulations have the same CFC-11 lower boundary conditions.





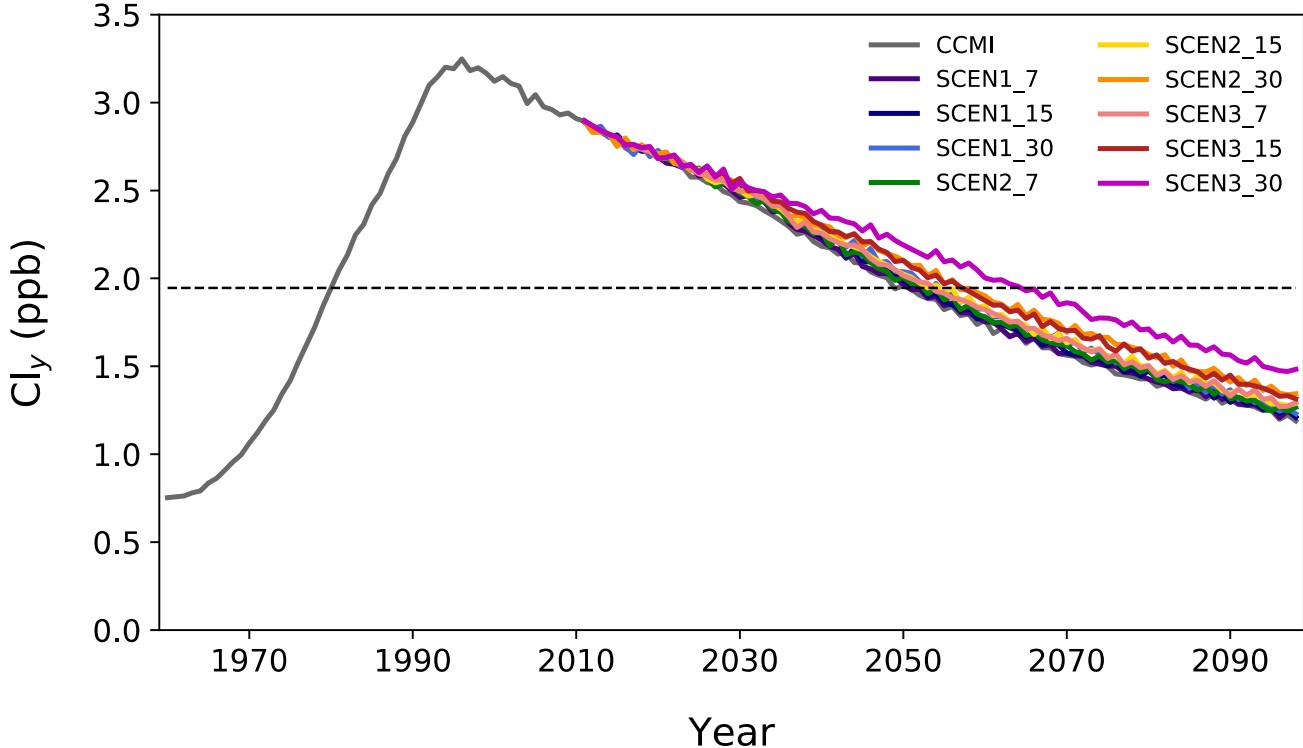

**Figure 3** Modelled annual mean 40 km inorganic chlorine (Cl$_y$) mixing ratios, averaged from 10°S-10°N, for the BASE and SCEN simulations. The dashed line represents the 1980 Cl$_y$ mixing ratio at 40 km in the BASE simulation, used as the value

5    for calculating Cl$_y$ return dates.




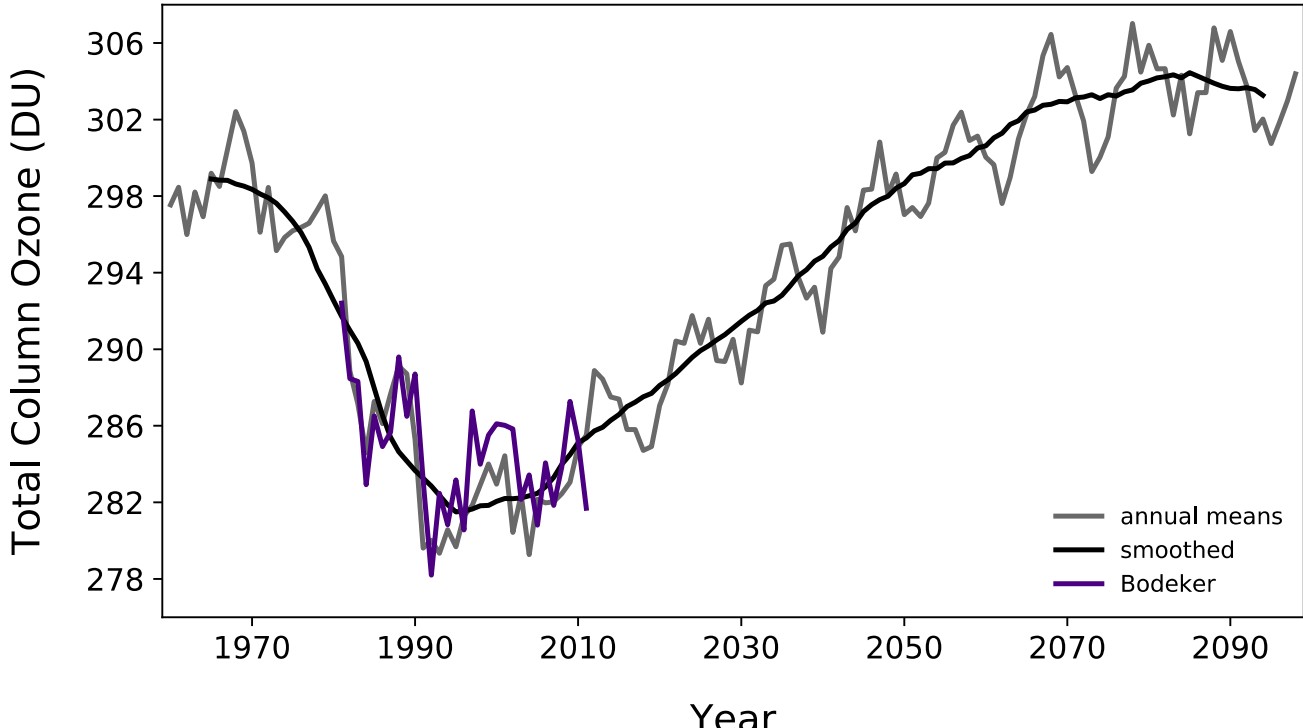

**Figure 4** Modelled annual mean TCO values (in DU), averaged from 60°S-60°N, for the BASE (grey line) and smoothed data (black line) using an 11-point boxcar smoothing to reduce both the effects of variability (following Dhomse et al., 2018). Also shown are TCO values from v2.8 of the Bodeker dataset (purple line; Bodeker et al., 2005).

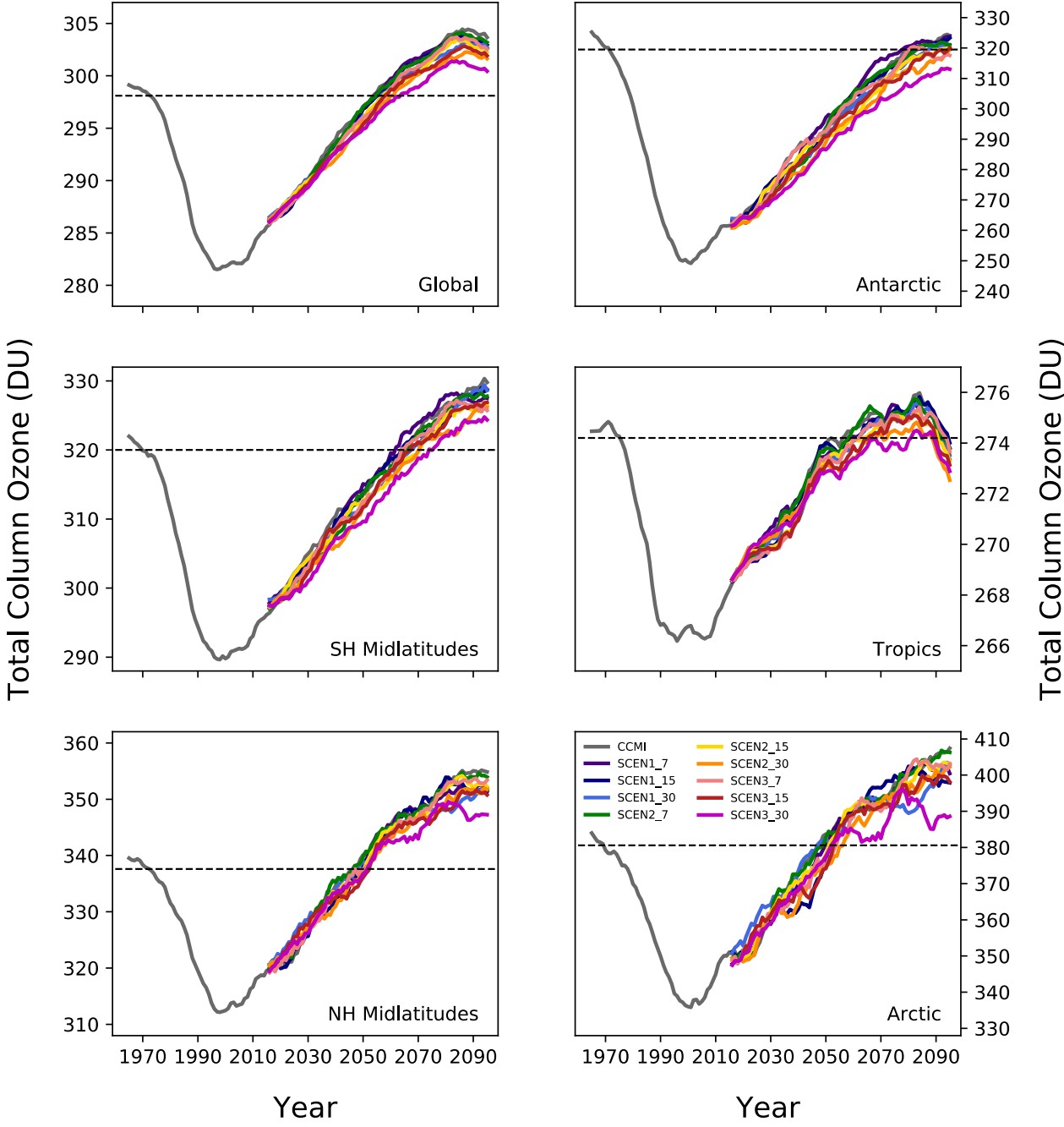

**Figure 5** Smoothed TCO (in DU) for the BASE and SCEN simulations, for the global (60°S-60°N) and regional column values. Dashed line denotes the 1960-1980 baseline period, used as the value for calculating TCO return dates.





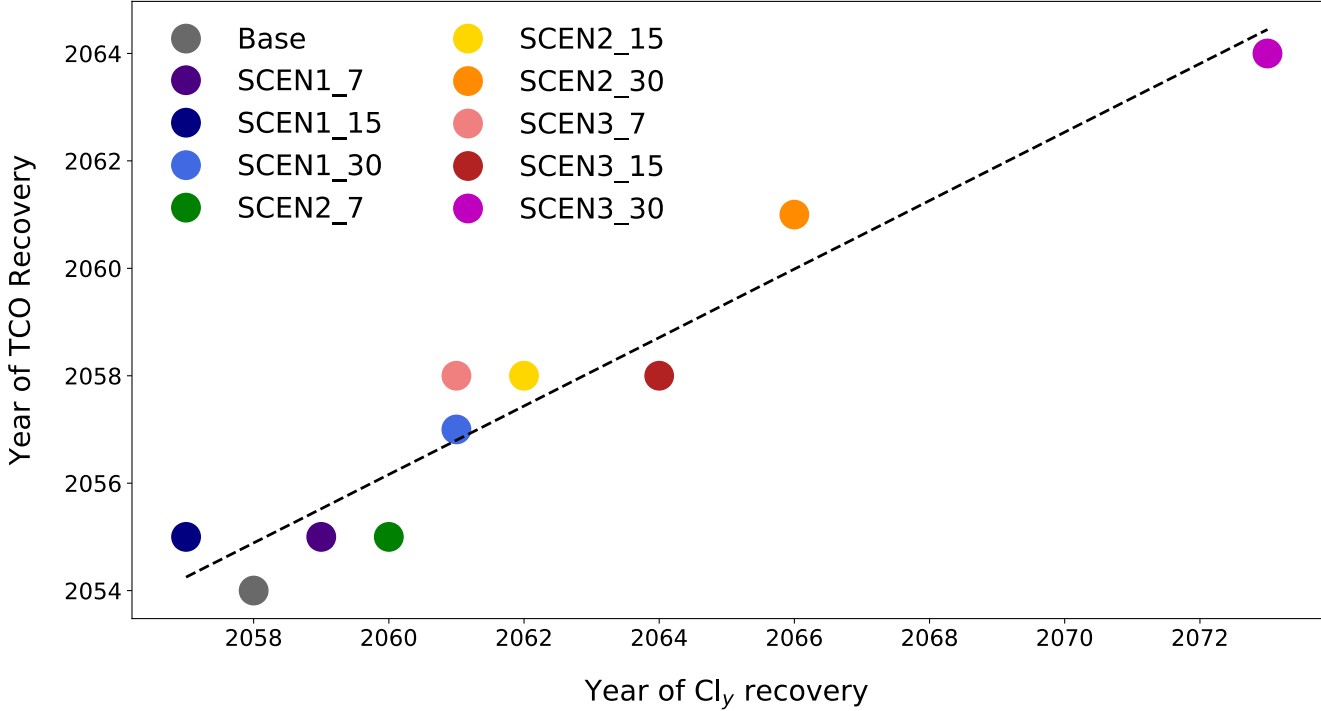

**Figure 6** Year of global TCO recovery (defined as the date at which TCO values return to the 1960-1980 average) for the BASE and SCEN simulations versus year of $Cl_y$ recovery (defined as the date at which $Cl_y$ mixing ratios at 40 km, averaged from 10°S-10°N, return to the 1980 BASE value). The dashed line gives the linear fit through the points.



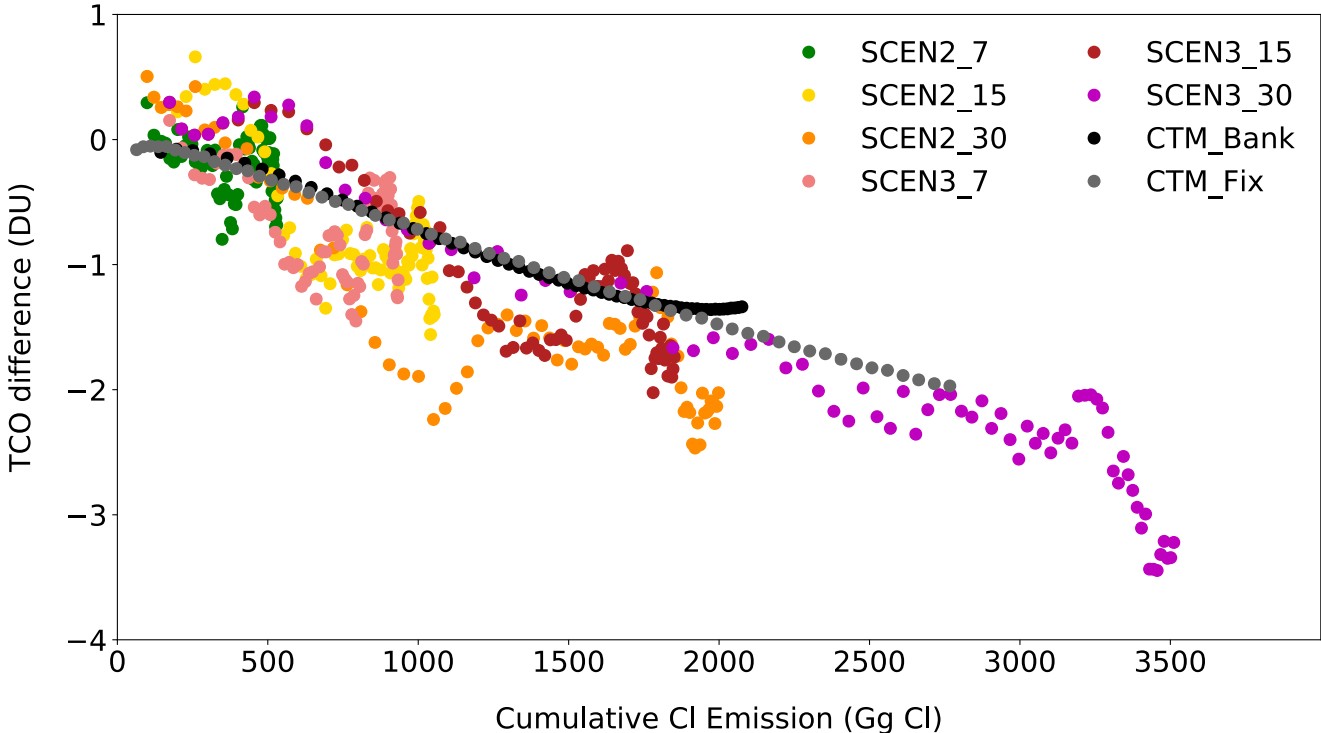

**Figure 7** Annual mean TCO difference (in DU) for the UM-UKCA SCEN2 and SCEN3 simulations with respect to the BASE simulation, averaged over 60∘S-60∘N vs cumulative emissions (Gg Cl) from 2012 to that year. Grey and black points show the same, but are for the CTM_Fix and CTM_Bank simulations performed with the TOMCAT CTM. For these simulations, TCO differences are calculated with respect to the CTM_C baseline simulation. TOMCAT simulations described in Dhomse et al. (2019).



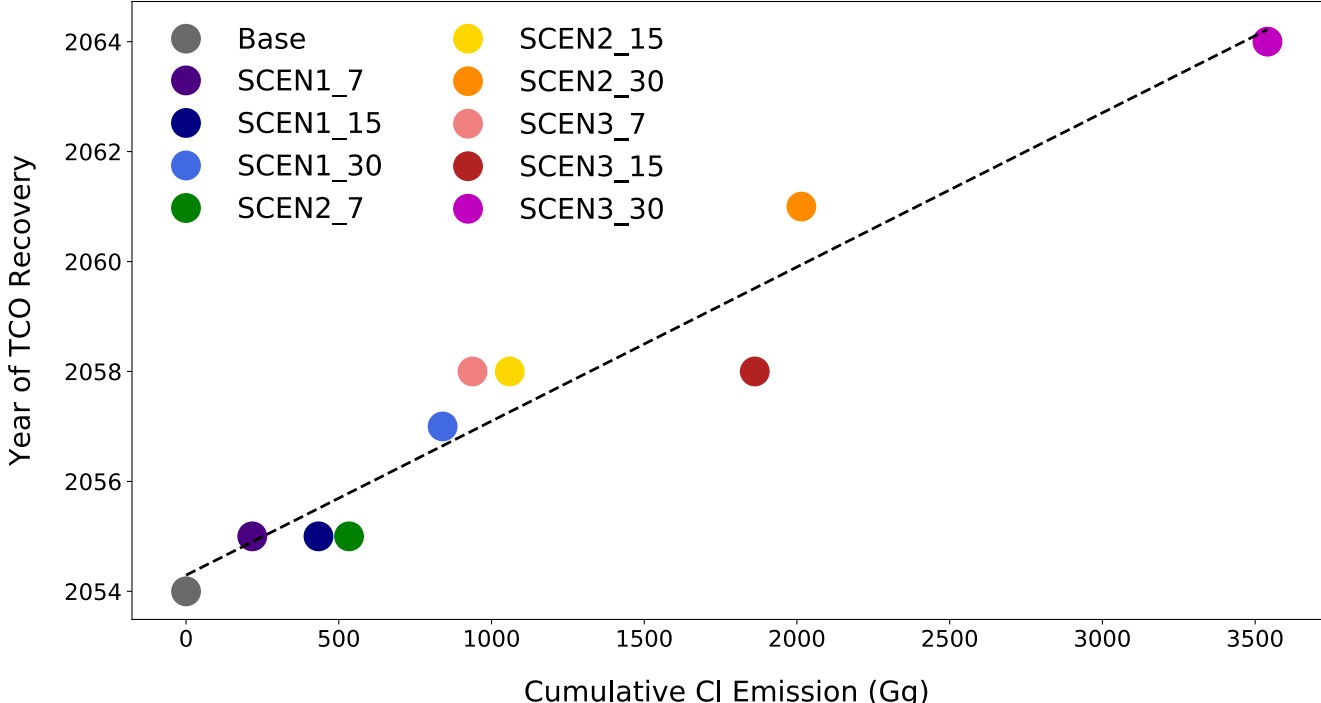

**Figure 8** Year of global TCO recovery (defined as the date at which TCO values return to the BASE 1960-1980 average) for the BASE and SCEN simulations vs cumulative additional emissions (Gg Cl) from 2012 to the end of the simulation. The dashed line gives the linear fit through the points.

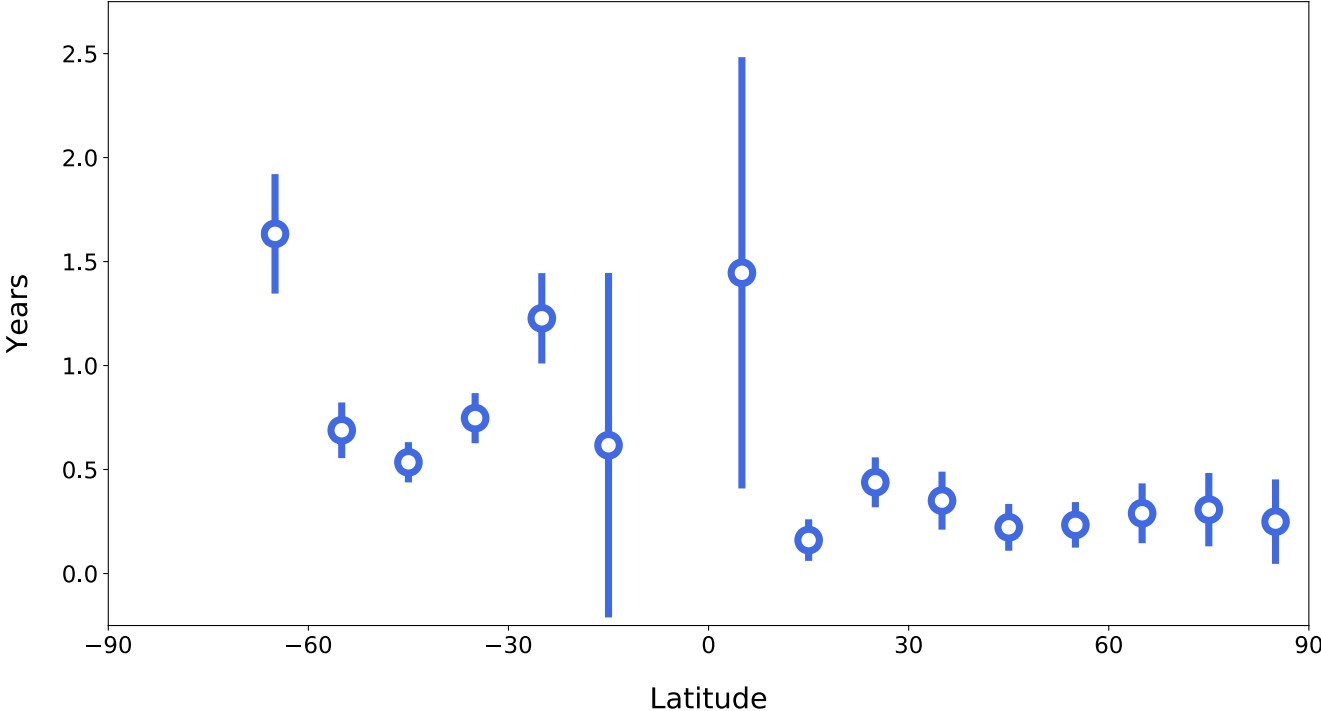

**Figure 9** Delay in timing of TCO recovery per 200 Gg Cl emissions for 10-degree latitude bins. Uncertainty bars represent the standard error of the estimate (calculation method provided in text). Note that no values are given for 90°S-80°S, 80°S-70°S and 10°S-0°N as in these simulations recovery does not occur by the end of the model simulations.