# Peer review of "Modelling the potential impacts of the recent, unexpected increase in CFC-11 emissions on total column ozone recovery"

_Atmospheric Chemistry and Physics, 2019_

## Referee Comment (RC1) · Anonymous Referee #2 · 30 Oct 2019

The manuscript presents a modelling investigation of the impact of recent increase in the emission of one of the major ozone depleting substance banned by the Montreal Protocol, the CFC-11. From the recently published evidence of recent emission increase, authors elaborate several scenarios for future emissions of CFC-11, with consequences on future levels of total inorganic chlorine and total ozone column throughout the 21st century. One interest of the study is that scenario independent metrics are explored to evaluate the relationship between ODS emissions and the timing of total ozone column recovery. The paper is well written and reference to previous work is adequate. It is suitable for publication in ACP provided that following comments and recommendations are considered in the revised version.

[Figure]

Major comments

1. Description of the model is limited and relates mainly on references. Very little information is provided on main features of tropospheric and stratospheric chemistry schemes. Are VSLS included? Since the model simulation output show a well-defined solar cycle effect, a brief description of the scheme used to integrate solar cycle variation would be valuable. Also, what is the lower boundary of the model?

2. The CFC-11 and CFC-12 emission scenarios are lacking explanation on the particular industrial activities they are based on. For example, what activities lead to emissive use of CFC-11? How realistic are the scenarios regarding industrial use of CFC-11? What is the reference for assuming equal emission of CFC-12 and CFC-11 in SCEN3 scenarios? A table summarizing the various scenarios would also be useful.

3. In Figure 3, the spread of return dates of inorganic chlorine Cly at 40 km is smaller than that of CFC-11 return dates in Figure 2. It would help to show total tropospheric emitted chlorine in order to understand this difference.

4. Figure 4: Comparison of baseline scenario with Bodeker dataset: it would be also interesting to compare with one of the datasets used in the last WMO Ozone Assessment (WMO, 2018), e.g. the GOME-SCIAMACHY-GOME-2 (GSG) product from the University of Bremen (Weber et al., 2011). I am not sure that the solar cycle is that pronounced in other data sets.

5. Figure 5: In order to evaluate the significance of the difference in return dates of the various scenarios, indication of uncertainty as indicated in page 7, l20-25 should be indicated in the figure. The largest spread in return dates is observed for Antarctica, the tropics and in the Arctic. Could the authors elaborate on the processes in the model simulation that explain this difference in spread of return dates? Is the simulated Brewer-Dobson circulation different in the various scenarios? Could that explain the difference?

[Figure]

6. Figure 6: the timing of Cly recovery is chosen from the date at which Cly at 40km averaged in the 10°N-10°S latitude range return to 1980 value. Is there a reason for choosing the 10°N-10°S latitude range?

7. Figure 7: the added value of additional results from TOMCAT simulations is not clear to me. Also in which time frame the additional TCO depletion is computed in the figure?

8. In Figure 8, and other correlation figures, the goodness of the fit is heavily influenced by the SCEN3-30 scenario, which contradicts in some way the statement on the scenario independent metrics. I would assume that for the results ranged in latitude bins, even larger error bars would be expected from the regression in the absence of that extreme scenario. Could the authors comment on that and on the significance of the delay in TCO recovery in some latitude bins in that case?

Specific comments

P3 l9-12: The text is somewhat confusing. It is not clear whether the increase in CFC-11 emissions is 35 Gg.y-1or 13 Gg.y-1 greater than expected from the Montreal Protocol.

P5 l19-20: explain relation between 1Gg CFC-11 and Gg Cl.

P13 l20-22: It is not clear why 200 Gg of CFC-11 emitted in 2020 would not have the same effect on ozone recovery as the same emitted in 2080.

In the text, some exponents are not well written, e.g. r2 values for the goodness of the fits.

---

## Referee Comment (RC2) · Anonymous Referee #3 · 18 Jan 2020

The paper reports on the impact of recent emissions of CFC-11 on the ozone layer, first documented in Montzka et al., and most recently, in Rigby et al. 2019. The paper explores a number of possible scenarios of CFC-11 emissions, and concludes that if emissions are allowed to continue into the future, significant delays in the recovery of the ozone layer are expected (by a decade or more in the global TCO, and even more in the Antarctic stratosphere). The subject of the paper is of high relevance and interest for the readership at ACP, and for the wider community. The paper is well written and the conclusions of the paper are supported by the data.

There are nonetheless a few issues in the paper, which need to be addressed in order

to make the paper suitable for publication. One of them is the missing discussion of another paper, which recently came out on the same subject (and in this same journal), namely Dameris et al., ACP 2019 (https://doi.org/10.5194/acp-19-13759-2019). I believe this paper is not discussed nor mentioned in the text because it came out quite recently, so the authors did not see it. Dameris et al., similar to this study, investigated the impact of two CFC-11 emission scenarios on ozone projections, and found that a substantial delay of ozone recovery by up to 20 years could occur due to unabated CFC11 emissions. Are the results in this paper consistent with their conclusions (bearing in mind their differences in the CFC11 scenarios)? There are also some issues in the missing discussion & quantification of the chemical mechanism leading to the delayed ozone recovery. Dameris et al. 2019 found that actually, some catalytic cycles are slowed down (e.g. Ox and NOx), leading to ozone increases under increased CFC11 emissions (which compensate the ozone depletion from enhanced ClOx cycles). For example, how important are ClOx cycles, relative to the other depleting cycles? In addition, the authors tend to only cite their own past papers, neglecting the much more vast body of literature on e.g., BDC and its impacts on ozone, and multi-model comparisons. Please cite the first papers that studies these problems, and not only your own ones; see specific comments below. The paper would benefit from addressing these issues.

SPECIFIC COMMENTS

- Can the present results be compared and discussed in the context of the recent paper by Dameris et al., 2019?

- Can the authors please try to pick a more descriptive experiment tag? SCENx-y is quite cumbersome and does not convey the basic information of the experiments setup. For example, how about "SCEN35" and "SCEN90" so that the reader immediately knows the total emission in terms of mass of CFC11... or something similar? The authors need to constantly remind the reader what each tag means, when they wish to emphasize some key result. This could be avoided by picking a better experiment tag.

- Could the authors look at the ClOx and other catalytic cycles, so that the reader can get a glimpse into the key chemical mechanism, and its dependence on height (without just simply assuming that the Molina and Rowland (1974) cycle is enhanced...? While I don't expect the authors to perform a full ozone budget analysis for each of the ensembles, showing the relative change in ozone production rates in a few key experiments would improve the paper.

MINOR COMMENTS

L20 page 2: Ball et al., ACP 2018 ( https://doi.org/10.5194/acp-18-1379-2018) is another key paper that should be cited here.

L5 page 5: why is the lifetime fixed? Shouldn't it be a strong function of altitude, and depend on the photolysis rate (which in turn is a function of the actinic flux)?

L12 page 7: again, Salby et al., 2009, and more recently Ball et al. 2018, also discussed the impact of variability on the detection of a recovery. So, these papers should be cited, too.

L15 page 7: how about the technique used in Eyring et al. 2013 and in all the WMO reports (the TSAM method)? This method seems more customary and it should be at least mentioned what the differences between this paper's method and that one is.

L31 page 8: this is probably subject to a lot of variability and running multiple ensembles may wash out this little increase TCO values. Or perhaps, there may be some compensating effect from NOx and Ox cycles. It would be good if the authors could look at the production rates (see specific comment above).

L6 page 9: the imapct of BDC accelleration on TCO has been studied in more studies than just these ones. Most recently, this has been shown in Chiodo et al., J.Clim. 2018 (DOI:10.1175/JCLI-D-17-0492.1 ). So, at the very least, add this paper to the references here.

L13 page 9; this has been first studied in Oman et al., 2009

(https://doi.org/10.1029/2010JD014362), so this paper should be cited, too.

L4-16 page 10: I don't get what the scenario-independent aspect would be, in this context. Cly clearly depends on the CFC emission scenario so there is no scenario independency here.

- The paper by Stolarski et al. 2012 (https://doi.org/10.1029/2012JD017456) shows that the sensitivity of ozone to temperature decreases as chlorine increases in the stratosphere. Hence, it should increase at a slower rate compared to the reference scenario, as chlorine decreases at a slower rate. This may impact the response of ozone to the upper stratospheric cooling, as a function of much Cly we have in the stratosphere. Could this explain some of the non-linearities found by the authors?

L8 page 12: other papers have reported this in the past, a lot more than just Keeeble et al., 2017, so they should be cited. At the very least, Butchart et al., 2014 should be cited here.

L20 page 13: where should this non-linearity come from (i.e. the same CFC11 change in 2080 having a different effect than in 2020) ? The Cl + O3 reaction rate changing because of O3 background concentrations? (the Cl change would be the same!)

---

## Author Comment (AC1) · 9 Apr 2020

We thank the reviewer for their time and comments. Please find our responses to each comment below - original reviewer comments in bold, author responses beneath.

**Referee #2**

**The manuscript presents a modelling investigation of the impact of recent increase in the emission of one of the major ozone depleting substance banned by the Montreal Protocol, the CFC-11. From the recently published evidence of recent emission increase, authors elaborate several scenarios for future emissions of CFC-11, with consequences on future levels of total inorganic chlorine and total ozone column throughout the 21st century. One interest of the study is that scenario independent metrics are explored to evaluate the relationship between ODS emissions and the timing of total ozone column recovery. The paper is well written and reference to previous work is adequate. It is suitable for publication in ACP provided that following comments and recommendations are considered in the revised version.**

**Major comments**

**1. Description of the model is limited and relates mainly on references. Very little information is provided on main features of tropospheric and stratospheric chemistry schemes. Are VSLS included? Since the model simulation output show a well-defined solar cycle effect, a brief description of the scheme used to integrate solar cycle variation would be valuable. Also, what is the lower boundary of the model?**

The model extends from the surface to 84 km. VSLS are included in the chemistry with prescribed, time evolving surface mixing ratios. Further information relating to the model configuration used for this study have been added to the manuscript in section 2. The model description section now reads:

"To explore the impacts of potential future CFC-11 emission scenarios on TCO return dates, a total of 10 transient simulations were performed using version 7.3 of the HadGEM3-A configuration of the Met Office's Unified Model (Hewitt et al., 2011) coupled with the United Kingdom Chemistry and Aerosol scheme (hereafter referred to as UM-UKCA). This configuration of the model has a horizontal resolution of 2.5° latitude × 3.75° longitude, and 60 vertical levels following a hybrid sigma-geometric height coordinate, extending from the surface to a model top at 84 km. The chemical scheme is an expansion of the scheme presented in Morgenstern et al. (2009) in which halogen source gases are considered explicitly and the effects of the solar cycle are considered, as described in Bednarz et al. (2016), and includes 45 chemical species, 118 bimolecular reactions, 17 termolecular reactions, 41 photolysis and 5 heterogeneous reactions occurring on the surfaces of polar stratospheric clouds and sulphate aerosols. This chemistry scheme provides a detailed treatment of stratospheric chemistry including the $O_x$, $ClO_x$, $BrO_x$, $HO_x$ and $NO_x$ catalytic cycles, and a simplified tropospheric scheme including the oxidation of a limited number of organic species ($CH_4$, CO, $CH_3O_2$, $CH_3OOH$, HCHO) alongside detailed $HO_x$ and $NO_x$ chemistry. The chemical tracers $O_3$, $CH_4$, $N_2O$, CFC-11, CFC-12, CFC-113, and HCFC-22 are all interactive with the radiation scheme. The halogenated source gases CFC-11, CFC-12, CFC-113, HCFC-22, halon-1211, halon-1301, $CH_3Br$, $CH_3Cl$, $CCl_4$, $CH_2Br_2$, and $CHBr_3$ are considered explicitly, the concentration of each prescribed at the surface as a time evolving lower boundary conditions (LBC).
The version of UM-UKCA used in this study is an atmosphere-only configuration, with each simulation using the same prescribed sea surface temperatures (SSTs) and sea ice fields taken

from a parent coupled atmosphere-ocean HadGEM2-ES integration. The configuration of the model used for this study includes the effects of the 11-year solar cycle in both the radiation and photolysis schemes. The top of atmosphere solar flux follows historical observations from 1960 to 2009, after which a repeating solar cycle is imposed which is an amplitude equivalent to the observed cycle 23 (as detailed in Bednarz et al., 2016). Further information on the model configuration used for this study is provided in Keeble et al. (2018). Except for CFC-11 and CFC-12 LBCs, all other chemical forcings in the simulations follow the experimental design of the WCRP/SPARC CCMI REF-C2 experiment (Eyring et al., 2013), which adopts the RCP6.0 scenario for future GHG and ODS emissions. A baseline experiment performed using CFC-11 and CFC-12 LBCs provided by the WMO (2014) A1 scenario was run from 1960 to 2099. A further 9 simulations were performed, running from 2012 to 2099, using a range of CFC-11 and CFC-12 LBCs, which were designed to cover a large but plausible range of potential future CFC emissions scenarios given the associated uncertainties."

**2. The CFC-11 and CFC-12 emission scenarios are lacking explanation on the particular industrial activities they are based on. For example, what activities lead to emissive use of CFC-11? How realistic are the scenarios regarding industrial use of CFC-11? What is the reference for assuming equal emission of CFC-12 and CFC-11 in SCEN3 scenarios? A table summarizing the various scenarios would also be useful.**

The scenarios explored in this study are not based on particular industrial activities, but instead are attempts to bound the potential impacts of recently reported CFC-11 emissions by exploring reasonable assumptions about future CFC-11 emissions and exploring the sensitivities to these assumptions. The direct emissions scenarios are not linked to any industrial activity, but instead explore the scenario in which the observed emissions reflect all the production (arguably the best-case scenario as it means there are no banks produced). This would be the case if the observed CFC-11 was being produced as a by-product in some other chemical manufacturing pathway and simply released. The most widely suggested use of recent CFC-11 production is as a blowing agent for foam insulation, so that the emissions represent some fraction of the total production, with the remainder building up in a newly created bank. In this case, we assume that for every 15 Gg directly emitted, 75 Gg enters the bank, i.e. 16.7% is released immediately, which is consistent with the 15% assumed by Dhomse et al. (2019) based on the study of Ashford et al., 2004. The reference for the ratio of co-production of CFC-12 is the Technology and Economic Assessment Panel. 2018 Assessment Report ([http://conf.montreal-protocol.org/meeting/oewg/oewg-41/presession/Background-Documents/TEAP_2018_Assessment_Report.pdf](http://conf.montreal-protocol.org/meeting/oewg/oewg-41/presession/Background-Documents/TEAP_2018_Assessment_Report.pdf)), which states that pure production of CFC-11 is difficult to achieve, with many plants producing CFC-11 and CFC-12 in ratio of 30:70 either way. 50:50 was chosen as the middle of this range. The TEAP assessment has been added as a reference to this section of the manuscript.

**3. In Figure 3, the spread of return dates of inorganic chlorine Cly at 40 km is smaller than that of CFC-11 return dates in Figure 2. It would help to show total tropospheric emitted chlorine in order to understand this difference.**

It is perhaps surprising that the spread in Cly at 40 km is smaller than the spread in CFC-11 return dates. We have further explored this, and it is not something peculiar at 40 km but is also true for a range of altitudes and latitudinal averages. It perhaps reflects the fact that CFC-11 is itself only a fraction of the total Cly, and that for many of the scenarios explored here CFC-11 mixing ratios return to their 1980s values before Cly mixing ratios, suggesting other halogenated source gases are controlling earliest return date of Cly.

We feel that, as the UM-UKCA model uses prescribed lower boundary concentrations, rather than emissions of CFC-11 and other halogenated source gases, showing the total tropospheric emitted chlorine may be misleading.

**4. Figure 4: Comparison of baseline scenario with Bodeker dataset: it would be also interesting to compare with one of the datasets used in the last WMO Ozone Assessment (WMO, 2018), e.g. the GOME-SCIAMACHY-GOME-2 (GSG) product from the University of Bremen (Weber et al., 2011). I am not sure that the solar cycle is that pronounced in other data sets.**

We have added the GOME-SCIAMACHY-GOME-2 data to the figure. There is some difference between the two products, but overall the model agrees well with both datasets.

[Figure]

**5. Figure 5: In order to evaluate the significance of the difference in return dates of the various scenarios, indication of uncertainty as indicated in page 7, l20-25 should be indicated in the figure. The largest spread in return dates is observed for Antarctica, the tropics and in the Arctic. Could the authors elaborate on the processes in the model simulation that explain this difference in spread of return dates? Is the simulated Brewer-Dobson circulation different in the various scenarios? Could that explain the difference?**

We are reticent to put the spread of return dates onto Figure 5 as they come from an ensemble of simulations using different halogenated source gas LBCs to those explored in this study. While we include results from this ensemble in Table 1 as a rough guide for uncertainty estimates, we make clear the caveats associated with these ranges in the text, and so including them on the figure may be misleading. As the reviewer states, large uncertainties are seen in the Arctic and tropics. In the Arctic this is the result of large interannual variability in TCO values associated with dynamical variations, while in the tropics the chemical ozone depletion signal is small and TCO variability is dominated by features such as the solar cycle, QBO and ENSO. In all scenarios, the BDC responds to the modelled ozone depletion, further contributing to the variability. Estimating uncertainties in the Antarctic was not possible as the year of TCO recovery occurs after 2080 for the latitude range 90S-60S, and this is denoted by the '?' in the table.

**6. Figure 6: the timing of Cly recovery is chosen from the date at which Cly at 40km averaged in the 10◦N-10◦S latitude range return to 1980 value. Is there a reason for choosing the 10◦N-10◦S latitude range?**

The 10°S-10°N latitudinal range is chosen as that is within the tropical pipe, in which air is predominantly moved vertically with limited horizontal mixing. This is necessary because Cly varies latitudinally with age of air. This has been added to the discussion of the figure, alongside references to Waugh 1996 and Neu and Plumb, 1999.

**7. Figure 7: the added value of additional results from TOMCAT simulations is not clear to me. Also in which time frame the additional TCO depletion is computed in the figure?**

The results from the TOMCAT model were included as consistency in the TCO difference vs Cl emission between it and the UM-UKCA model lends confidence to the rest of UKCA results, particularly given the differences in both the model configurations and the choice of CFC-11 emissions scenarios. It also suggests that the dominant driver of the response is the chemistry, and not circulation feedbacks, as the TOMCAT model is run with constant meteorological fields.

Figure 7 uses data from 2012-2100 for the UM-UKCA simulations, and 2012-2080 for the TOMCAT simulations. Each point on the figure represents one modelled year for a given scenario, and plots the TCO difference in that year between that scenario and base simulation against the cumulative Cl emissions reached in that scenario by that year. This has been clarified in the text.

**8. In Figure 8, and other correlation figures, the goodness of the fit is heavily influenced by the SCEN3-30 scenario, which contradicts in some way the statement on the scenario independent metrics. I would assume that for the results ranged in latitude bins, even larger error bars would be expected from the regression in the absence of that extreme scenario. Could the authors comment on that and on the significance of the delay in TCO recovery in some latitude bins in that case?**

The goodness of the fit is to some extent dependent on the SCEN3-30 point, but even when it is removed there still exists a strong correlation for the data in both Figure 6 (year of TCO recovery vs Cly recovery) and Figure 8 (year of TCO recovery vs cumulative Cl emissions). For Figure 6, when the SCEN3-30 experiment is ignored, the $r^2$ changes from 0.92 to 0.85, while the gradient changes from 0.64 to 0.72, while for Figure 8, when the SCEN3-30 experiment is ignored, the $r^2$ changes from 0.93 to 0.85, while the gradient changes from 0.56 to 0.59. This suggests that the relationships identified in the paper are not heavily influenced by the SCEN3-30 scenario.

This is true for the results in latitude bins as well. Below is a version of Figure 9 from the paper, with the blue points the same as those presented and using all the experiments, while the red points are the same calculations but ignoring the SCEN3-30 scenario. While the uncertainty estimates are changed slightly, and the delay in the Arctic is slightly longer per emission of 200 Gg Cl, there is no significant change, again showing that the SCEN3-30 scenario alone does not heavily influence the conclusions of the paper. We have left the original Figure 8 the same in the manuscript as it uses all the available data.

[Figure]

**Specific comments:**

**P3 l9-12: The text is somewhat confusing. It is not clear whether the increase in CFC-11 emissions is 35 Gg.y-1or 13 Gg.y-1 greater than expected from the Montreal Protocol.**

Observational evidence indicates that CFC-11 emissions increased by ~13 Gg yr-1 after around 2012, and that a large fraction of this increase came from East Asia. This increase, in conjunction with the expected decline in global CFC-11 emissions which would have resulted from full compliance with the Montreal Protocol, has resulted in global CFC-11 emissions being ~35 Gg yr-1 higher than expected in 2019. This clarification has been added to the text, and the paragraph now reads:

"Against this background, Montzka et al. (2018) showed that the atmospheric abundance of one of the major chlorine-carrying CFCs, CFC-11, is not declining as expected under full compliance with the Montreal Protocol. Using the NOAA network of ground-based observations, they demonstrated clearly that the rate of decline of CFC-11 in the atmosphere between 2015-2017 was about 50% slower than that observed during 2002–2012 and was also much slower than had been projected by WMO (2014). They inferred that emissions of CFC-11 had been approximately constant at ~55 Gg yr-1 between 2002 and 2012 and had then risen after 2012 by 13 Gg yr-1 to ~68 Gg yr-1. Montzka et al. (2018) argued that this increase could not be explained by increased release from pre-existing banks. Instead, they suggested that production of CFC-11 in east Asia, which is inconsistent with full compliance of the Montreal Protocol, was the likely cause. Using inverse modelling, Rigby et al. (2019) have since shown that the increase in CFC-11 emissions from eastern China between 2008-2012 and 2014-2017 is ~7 Gg yr-1, corresponding to approximately 40-60% of the global emission increase identified by Montzka et al. (2018) during that period. The increase in CFC-11 emissions after 2012, in conjunction with the expected decline in global CFC-11 emissions which would have resulted from full compliance with the Montreal Protocol, has resulted in global CFC-11 emissions being ~35 Gg yr-1 greater than anticipated by the WMO (2014) A1 scenario in 2019."

**P5 l19-20: explain relation between 1Gg CFC-11 and Gg Cl.**

The relationship between the two is 1Gg CFC-11 = 0.77 Gg Cl. This is based on the fact that the molar mass of CFC-11 is 137.37 g/mol and contains 3 molecules of Cl (molar mass = 35.453, multiplied by 3 is 106.359). Therefore, every Gg of CFC-11 contains 106.359/137.37 = 0.77 Gg Cl.

**P13 l20-22: It is not clear why 200 Gg of CFC-11 emitted in 2020 would not have the same effect on ozone recovery as the same emitted in 2080.**

This statement arises from two considerations. The first is related to the non-linear coupling between chlorine emission, background stratospheric composition and climate. The same emission of CFC-11 has a different relative impact on stratospheric ozone for different stratospheric temperatures arising from the different temperature dependences of ozone depleting reactions. Further, changes to CH4 and N2O lower boundary conditions also change the relative proportions of the HOx, NOx and ClOx catalytic cycles, and the partitioning of stratospheric chlorine. Studies by Banerjee et al. (2016), and Keeble et al. (2017) have explored this non-linear response of column ozone changes to ODS changes under different future climates in the UKCA model, and these results are consistent with the original findings of Haigh and Pyle (1982) and recent studies from other groups (e.g. Fleming et al., 2011; Portmann et al., 2012; Meul et al., 2015). The second is that 200 Gg of CFC-11 emitted into the atmosphere after the year of ozone recovery when chlorine mixing ratios have already been significantly reduced may not reduce column ozone values below the 1980s/1960s value, and so while this leads to depletion of ozone it does not influence return dates. The text in the manuscript has been modified to reflect these points, and the paragraph now reads:

"It is not expected that an emission of CFC-11 emitted in 2020 would have the same impact on ozone return dates as the same emission of CFC-11 emitted in, for example, 2080. This is due in part to the different background stratospheric temperatures, circulation, and sinks of active chlorine (e.g. the conversion of ClOx to HCl through reaction with CH4) at different times throughout the 21st century. Furthermore, any additional chlorine emissions which occur after TCO has returned to its 1960-1980 mean value might not deplete ozone below this value, and so would not affect the return date.

**In the text, some exponents are not well written, e.g. r2 values for the goodness of the fits.**

These have been corrected.

---

## Author Comment (AC2) · 9 Apr 2020

We thank the reviewer for their time and comments. Please find our responses to each comment below - original reviewer comments in bold, author responses beneath.

**Anonymous Referee #3**

**The paper reports on the impact of recent emissions of CFC-11 on the ozone layer, first documented in Montzka et al., and most recently, in Rigby et al. 2019. The paper explores a number of possible scenarios of CFC-11 emissions, and concludes that if emissions are allowed to continue into the future, significant delays in the recovery of the ozone layer are expected (by a decade or more in the global TCO, and even more in the Antarctic stratosphere). The subject of the paper is of high relevance and interest for the readership at ACP, and for the wider community. The paper is well written and the conclusions of the paper are supported by the data. There are nonetheless a few issues in the paper, which need to be addressed in order to make the paper suitable for publication. One of them is the missing discussion of another paper, which recently came out on the same subject (and in this same journal), namely Dameris et al., ACP 2019 (https://doi.org/10.5194/acp-19-13759-2019). I believe this paper is not discussed nor mentioned in the text because it came out quite recently, so the authors did not see it. Dameris et al., similar to this study, investigated the impact of two CFC-11 emission scenarios on ozone projections, and found that a substantial delay of ozone recovery by up to 20 years could occur due to unabated CFC11 emissions. Are the results in this paper consistent with their conclusions (bearing in mind their differences in the CFC11 scenarios)? There are also some issues in the missing discussion & quantification of the chemical mechanism leading to the delayed ozone recovery. Dameris et al. 2019 found that actually, some catalytic cycles are slowed down (e.g. Ox and NOx), leading to ozone increases under increased CFC11 emissions (which compensate the ozone depletion from enhanced ClOx cycles). For example, how important are ClOx cycles, relative to the other depleting cycles? In addition, the authors tend to only cite their own past papers, neglecting the much more vast body of literature on e.g., BDC and its impacts on ozone, and multi-model comparisons. Please cite the first papers that studies these problems, and not only your own ones; see specific comments below. The paper would benefit from addressing these issues.**

We thank the reviewer for their time and comments. Please find our detailed responses below. The referee rightly indicates that more comprehensive referencing is requited in some areas – we have been happy to do this.

**SPECIFIC COMMENTS**

**- Can the present results be compared and discussed in the context of the recent paper by Dameris et al., 2019?**

The Dameris et al. results have now been discussed in relation to the results presented in this manuscript in the introduction and discussion sections.

**- Can the authors please try to pick a more descriptive experiment tag? SCENx-y is quite cumbersome and does not convey the basic information of the experiments setup. For example, how about "SCEN35" and "SCEN90" so that the reader immediately knows the total emission in terms of mass of CFC11... or something similar? The authors need**

**to constantly remind the reader what each tag means, when they wish to emphasize some key result. This could be avoided by picking a better experiment tag.**

We feel that the scenario names chosen best reflect the emissions scenarios being considered as they include both the scenario type (1 = direct emission, 2 = formation of a bank, 3 = co-production of CFC-12) and also the duration of the assumed CFC-11 production, in years, in as succinct a manner as possible. It is not possible to use only the total emission of CFC-11 to name the scenarios as the emissions change through time for the scenarios which assume a bank is produced, and nor would it reflect the duration of the production.

**- Could the authors look at the ClOx and other catalytic cycles, so that the reader can get a glimpse into the key chemical mechanism, and its dependence on height (without just simply assuming that the Molina and Rowland (1974) cycle is enhanced...? While I don't expect the authors to perform a full ozone budget analysis for each of the ensembles, showing the relative change in ozone production rates in a few key experiments would improve the paper.**

[Figure]

We agree with the reviewer that it is unlikely that only the ClOx catalysed ozone depleting cycles will be affected by the changes to the CFC-11 and CFC-12 lower boundary conditions. Above is a plot of the relative difference in global mean ozone depletion through the ClOx, BrOx, NOx, HOx and Ox catalytic cycles between the SCEN2_30 simulation and BASE (averaged from 2030-2040). As found in the Dameris study, increases to ozone destruction through the ClOx catalytic cycles are somewhat offset by decreases to ozone loss through other catalytic cycle. However, interpretation of the drivers of these changes are complicated. This is partly through the chemical coupling of the ClOx cycles with the NOx, HOx and BrOx cycles, and partly through changes to the Brewer-Dobson circulation affecting the oxidation of source gases. Diagnosing the changes to these cycles is further complicated by the need to account for the changes to O3 concentrations. Increases to ClO will deplete stratospheric ozone, and so we may expect, for example, NOx catalysed ozone depletion to slow as 1) there is extra formation of the ClONO2 reservoir, and 2) there is a lower concentration of O3.

Because of these reasons, we feel that addressing the question of which catalytic cycles are responsible for the TCO changes examined in the paper is beyond the scope of this study. It should be stated that we have not assumed in the text that only the Molina and Rowland cycles are affected but have added text to the discussion to include the points above and cited the Dameris et al. ozone budget results in the discussion section of the manuscript.

**MINOR COMMENTS**

**L20 page 2: Ball et al., ACP 2018 (https://doi.org/10.5194/acp-18-1379-2018) is another key paper that should be cited here.**

This citation has been added to the discussion.

**L5 page 5: why is the lifetime fixed? Shouldn't it be a strong function of altitude, and depend on the photolysis rate (which in turn is a function of the actinic flux)?**

While the local lifetime of CFC-11 is dependent on its destruction through photolysis, the 55 year lifetime is used to convert the emissions fluxes into a mixing ratio at the surface of the model, and so represents its bulk atmospheric lifetime (i.e. the global burden/global loss). Imagining a pulse emission of CFC-11, some is transported into the middle and upper stratosphere where it is rapidly destroyed, but the majority (by mass) remains in the troposphere and lower stratosphere, where its lifetime is much longer. A lifetime of 55 years accurately projects the decline in the mixing ratio observed at the surface, and hence its use in this study. The value of 55 years for the lifetime comes from Chipperfield et al., 2014 (referenced in the manuscript).

**L12 page 7: again, Salby et al., 2009, and more recently Ball et al. 2018, also discussed the impact of variability on the detection of a recovery. So, these papers should be cited, too.**

We have added the Ball citation to the manuscript. We could not find a Salby et al., 2009 publication, but have cited instead the Salby et al., 2011 paper which discusses the impact of variability on the detection of Antarctic ozone recovery (Rebound of Antarctic ozone, GRL).

**L15 page 7: how about the technique used in Eyring et al. 2013 and in all the WMO reports (the TSAM method)? This method seems more customary and it should be at least mentioned what the differences between this paper's method and that one is.**

This method has been added to the manuscript, along with the Scinocca et al. (2010) reference. The sentence now reads:

"To mitigate these impacts, the effects of natural processes (such as volcanic eruptions, the QBO, ENSO and solar cycle) can be removed from the data using statistical techniques (such as Multiple Linear Regression, e.g. Staehelin et al., 2001; WMO, 2007 or the Time series Additive Model, Scinocca et al., 2010), or the data can be smoothed by averaging across multiple years (e.g. Dhomse et al., 2018)."

**L31 page 8: this is probably subject to a lot of variability and running multiple ensembles may wash out this little increase TCO values. Or perhaps, there may be some compensating effect from NOx and Ox cycles. It would be good if the authors could look at the production rates (see specific comment above).**

As the reviewer states, these differences likely result from variability between the integrations and would likely be removed by the use of more ensemble members. However, it was decided it would be better to explore a larger spread in possible future CFC-11 production scenarios. Please see our comment above about analysing the ozone budget terms in these simulations.

**L6 page 9: the imapct of BDC accelleration on TCO has been studied in more studies than just these ones. Most recently, this has been shown in Chiodo et al., J.Clim. 2018 (DOI:10.1175/JCLI-D-17-0492.1). So, at the very least, add this paper to the references here.**

Additional references have been added to this discussion. The sentence now reads:
"The observed ozone loss in the tropics has been small and, furthermore, future changes in the tropics are driven both by reductions in the stratospheric abundance of halogens, which tend to increase ozone, and the strengthening of the Brewer-Dobson circulation, which tends to decrease column ozone (e.g. Oman et al., 2010; Eyring et al., 2013; Meul et al., 2014; Keeble et al., 2017; Chiodo et al., 2018)."

**L13 page 9; this has been first studied in Oman et al., 2009 (https://doi.org/10.1029/2010JD014362), so this paper should be cited, too.**

This reference has been added to the manuscript.

**L4-16 page 10: I don't get what the scenario-independent aspect would be, in this context. Cly clearly depends on the CFC emission scenario so there is no scenario independency here.**

By 'scenario independent' we mean one could use the linear relationships identified between the amount of chlorine emitted and the timing of TCO return to estimate the delay in TCO return date for any particular chlorine emission. The benefit of this extends to emissions not explored in this study (i.e. not one of the 10 scenarios investigated). As the reviewer states, the $Cl_y$ projection depends on the assumed CFC emission scenario, but it is not known what possible future emission scenario will be followed. Rather than argue that one of the scenarios explored in this study is more likely than the others, we have looked for ways to say 'if the emission of CFC-11 was x, then the delay would be y', which we believe would represent a positive benefit to policy and regulation.

**- The paper by Stolarski et al. 2012 (https://doi.org/10.1029/2012JD017456) shows that the sensitivity of ozone to temperature decreases as chlorine increases in the stratosphere. Hence, it should increase at a slower rate compared to the reference scenario, as chlorine decreases at a slower rate. This may impact the response of ozone to the upper stratospheric cooling, as a function of much Cly we have in the stratosphere. Could this explain some of the non-linearities found by the authors?**

The reviewer is correct to highlight this process as a potential cause for the non-linearities seen here. An additional factor is coupling between the chemistry and dynamics. However, despite these non-linear couplings, and the interannual variability inherent in CCMs, the linear relationships identified in this study between the amount of chlorine emitted and the delay in ozone recovery are robust, with high r2 values, suggesting that these factors a minor in the face of the dominant effect of changing stratospheric Cl and depletion of ozone.

**L8 page 12: other papers have reported this in the past, a lot more than just Keeeble et al., 2017, so they should be cited. At the very least, Butchart et al., 2014 should be cited here.**

The Butchart et al. and Meul et al., 2016 references have been added to the discussion.

**L20 page 13: where should this non-linearity come from (i.e. the same CFC11 change in 2080 having a different effect than in 2020) ? The Cl + O3 reaction rate changing because of O3 background concentrations? (the Cl change would be the same!)**

Please see our response to the same question from referee #2